Subject Areas:
psychology/developmental biology

Keywords:
developmental prosopagnosia, object recognition, face specificity, novel object memory test

Author for correspondence:
Regan Fry
e-mail: regan_fry@hms.harvard.edu

# Evidence for normal novel object recognition abilities in developmental prosopagnosia

Regan Fry[1,3], Jeremy Wilmer[4], Isabella Xie[5,6], Mieke Verfaellie[2,7] and Joseph DeGutis[1,3]

[1]Boston Attention and Learning Laboratory, and [2]Memory Disorders Research Center, VA Boston Healthcare System, Boston, MA, USA
[3]Department of Psychiatry, Harvard Medical School, Boston, MA, USA
[4]Department of Psychology, Wellesley College, Wellesley, MA, USA
[5]Washington University in St Louis, St Louis, MO, USA
[6]Harvard Decision Science Lab, Harvard Kennedy School, Cambridge, MA, USA
[7]Department of Psychiatry, Boston University School of Medicine, Boston, MA, USA

RF, 0000-0001-6002-626X

The issue of the face specificity of recognition deficits in developmental prosopagnosia (DP) is fundamental to the organization of high-level visual memory and has been increasingly debated in recent years. Previous DP investigations have found some evidence of object recognition impairments, but have almost exclusively used familiar objects (e.g. cars), where performance may depend on acquired object-specific experience and related visual expertise. An object recognition test not influenced by experience could provide a better, less contaminated measure of DPs' object recognition abilities. To investigate this, in the current study we tested 30 DPs and 30 matched controls on a novel object memory test (NOMT Ziggerins) and the Cambridge Face Memory Test (CFMT). DPs with severe impairment on the CFMT showed no differences in accuracy or reaction times compared with controls on the NOMT. We found similar results when comparing DPs with a larger sample of 274 web-based controls. Additional individual analyses demonstrated that the rate of object recognition impairment in DPs did not differ from the rate of impairment in either control group. Together, these results demonstrate unimpaired object recognition in DPs for a class of novel objects that serves as a powerful index for broader novel object recognition capacity.

# 1. Introduction

Developmental prosopagnosia (DP) is a neurodevelopmental disorder resulting in lifelong face recognition deficits in the absence of brain injury or co-occurring social, intellectual or visual impairments [1–3]. The controversy over whether DP is specific to faces or involves additional domain-general deficiencies has been raised a number of times in the last 20 years [4–6], particularly regarding the presence of co-occurring deficits in object recognition.[1] Evidence has shown that at least some DPs have highly face-specific deficits, such as the case of Edward, a DP with preserved familiar object recognition and recall despite profound impairments in face perception and recognition ([5]; for others, see [8,9]). These DP cases clearly show that face recognition can be dissociated from familiar object recognition, providing important evidence that they rely on distinct mechanisms.

Despite these clear cases of dissociation, studies suggest that face and familiar object recognition deficits in DPs may co-occur. DPs have shown higher familiar object recognition impairment rates than controls [6]. In a comprehensive meta-analysis of DP cases, Geskin & Behrmann [6] examined a wide range of familiar object categories and found that, of those studies with both accuracy and reaction time measures, 80.3% (191/238) of DPs experienced co-occurring object recognition deficits (either accuracy *or* reaction time $z$-score $< -1.7$). When including all DPs with accuracy data,[2] excluding DPs with only a reaction time impairment, and using a more standard $z < -2$ cut-off, the DP familiar object recognition impairment rate was still a substantial 22.0% (101/459), much higher than the 2.5% rate predicted in controls.

In studies of familiar object recognition, DPs have often shown reduced, though normal, group-level performance for object categories such as bicycles [10,11], shoes [12], shells [13] and flowers [14]. However, in a recent large study of 46 DPs, Gray *et al.* [15] found that, compared with controls, the DP group scored significantly worse (Cohen's $d = 0.49$) on the Cambridge Car Memory Test (CCMT; [15,16]). The CCMT is a sensitive and widely used test of familiar object recognition that matches the Cambridge Face Memory Test (CFMT; [17]) in format, and cars are a category of objects for which participants typically have very high levels of experience and expertise [18]. Similar-sized decrements in car recognition have also been observed in smaller samples of DPs (e.g. [19,20]; though not all, see [21]). Interestingly, Barton *et al.* [10] found that DPs and controls had similar car recognition performance on a long-term memory task assessing recognition of familiar cars, but when taking into account semantic car knowledge, DPs showed significantly reduced performance compared with controls. Although the long-term recognition task from Barton *et al.* [10] differs from shorter-term episodic memory tasks typically used to assess familiar object recognition, these results raise the interesting possibility that DPs' familiar object recognition impairments, when observed, may be due to a failure to benefit from experience with these objects rather than to a domain-general impairment in object recognition.

To test the hypothesis that DPs have an impairment that is specific to experience-dependent learning of familiar objects, we used a category of objects for which no-one has experience: novel objects. Novel objects are laboratory-created stimuli that have similar overall shape and can be discriminated based on parts that vary between individual cases (e.g. rounded versus rectangular projections, double- versus single-sided edges). Because novel objects are not influenced by category-specific experience or expertise, they are thought to better measure the domain-general visual capacity to learn and recognize objects across all categories [22]. Supporting this, Richler *et al.* [23] demonstrated that three versions of a novel object memory test (NOMT) all show more shared domain-general variance (average 23%) than tests using familiar objects (e.g. cars, horses, guns; average 10% shared variance). NOMTs are uniquely interesting in that one's performance is thought to reflect visual perception, learning and memory abilities without the moderating effects of prior experience [22,23]. Performing NOMTs with DPs could have important theoretical importance; if DPs show worse group-level performance than controls on NOMTs, this would be indicative of more general deficits in perception, encoding or retrieval of objects. Conversely, normal group-level performance would suggest that DPs' category-general object abilities are intact and that impairments in recognition of highly familiar

---

[1]The term 'recognition' has sometimes been used to refer to perceptual matching (e.g. the Benton Face Recognition Test) and other times to memory (e.g. Warrington Recognition Memory Test for Faces). In this paper, we use 'recognition' as it refers to memory, specifically 'the ability to identify information as having been encountered before' [7].

[2]We recalculated impairment based on accuracy alone because nearly all of these object recognition tests were given without instructions to perform as quickly as possible and also because it has yet to be shown that reaction time on any of these tests explains unique variance beyond accuracy in DP diagnosis or CFMT scores.

objects (e.g. cars) are more likely to be due to an inability to benefit from experience. One previous study tested DPs and controls using a novel object recognition memory test ('blue objects', [13]). As a group, DPs performed comparably to controls but mean accuracy was very low in both groups ($d' = 0.6$), suggestive of floor effects. This makes it challenging to rule out differences between DPs and controls. An aim of the current study was to better assess DPs versus control differences by using a novel object recognition memory test with a broad range of scores and better overall performance.

In the current study, we sought to characterize DPs' novel object recognition abilities by administering the NOMT Ziggerins and the similarly formatted CFMT to a sample of 30 DPs and 30 age- and gender-matched controls. The Ziggerins were chosen because they bear no resemblance to either faces or familiar objects, have high internal reliability (Cronbach's $\alpha = 0.89$; [23]), and are free from floor effects. Performance on Ziggerins also correlates highly with other measures of visual object recognition while dissociating from vocabulary and digit span, suggesting that general verbal ability does not significantly contribute to Ziggerins performance [23]. Further, we compared DP scores to 274 controls from a recent study by Richler *et al.* [23]. In addition to these group-level analyses, we employed case-based analyses to determine whether (i) individual DPs showed impairment on the NOMT Ziggerins, and (ii) any DPs showed a classical dissociation between object and face recognition [24]. By comparing the performance of DPs and typically developed (TD) controls on the CFMT and a NOMT at both the group and individual levels, we sought to characterize novel object recognition in DPs.

# 2. Material and methods

## 2.1. Participants

Participants were between the ages of 18 and 70 years old ($N = 60$). Developmental prosopagnosics were recruited from our database of previous DP participants in the Boston area, references from other research laboratories (Dr Matthew Peterson, Massachusetts Institute of Technology; Dr Brad Duchaine, Dartmouth College, www.faceblind.org), and individuals who responded to our advertisement on the Massachusetts Bay Transportation Authority subway system. Control subjects were recruited from both the Harvard Decision Science Laboratory in Cambridge, Massachusetts and through flyers distributed in the Boston area.

The DP group did not differ from the in-lab typically developed control group in either age ($M_{DP} = 38.50$, s.d.$_{DP} = 13.69$, $M_{TD} = 40.03$, s.d.$_{TD} = 11.73$, $p = 0.643$), gender (DP$_{Female} = 24$, TD$_{Female} = 19$, $t_{57} = 1.70$, $p = 0.095$), or education ($M_{DP} = 17.52$, s.d.$_{DP} = 2.57$, $M_{TD} = 16.17$, s.d.$_{TD} = 2.80$; $t_{57} = 1.93$, $p = 0.059$). Education was measured for the in-lab groups by years of formal education, with options ranging from 1st grade (coded as 1) to 4th year of graduate school (coded as 20). The DP group did not differ from the web controls in age ($M_{Web} = 36.78$, s.d.$_{Web} = 12.04$, $p = 0.464$); however, the DP group had a higher proportion of females ($N_{Female} = 24$, 80.0%) than the web control group ($N_{Female} = 160$, 58.2%; $t_{298} = 3.06$, $p = 0.004$). For results separated by gender, see electronic supplementary material, figure S1. The web controls also had a lower average education level ($M_{DP} = 4.52$, s.d.$_{DP} = 0.95$, $M_{TD} = 3.76$, s.d.$_{TD} = 1.04$; $t_{273} = 3.74$, $p < 0.001$). The web controls' education was measured on a scale of 1–5, where 1 = middle school, 2 = high school/secondary school, 3 = some college/university, 4 = bachelor's degree and 5 = graduate degree. For comparison between the two groups, DP education levels were re-coded to match this scale. While ideally an education-matched web control group is preferred, studies have shown that face recognition is independent from both education and IQ [25]. Similarly, education has previously shown not to predict any additional variance in general memory performance outside of age [26].

## 2.2. DP and control qualifications

Developmental prosopagnosics were screened using the 20-Item Prosopagnosia Index (PI-20; [27]), a famous faces memory test (FFMT) and the CFMT [17]. To qualify as a DP, participants had to have scored above 64 on the PI-20, more than two standard deviations below the control mean on the FFMT and 44 or lower out of 72 on the original CFMT [28,29]. Seven potential DPs were excluded on the basis of CFMT score. The PI-20 and FFMT were given as a pre-screening measure, while the CFMT was included as part of the battery. All participants had normal or corrected-to-normal vision, and had to have scored within the normal range on the Leuven Perceptual Organization Screening Test (L-POST; [30]) to rule out other causes of poor face recognition. No DPs were excluded on the basis of L-POST score. In-lab controls had similar criteria with the exception that they must *not* report any lifelong face recognition issues, score greater than 64 on the PI-20, or score lower than 45 on the

CFMT. Control subjects were only invited to participate in the entire study if they met the above criteria. Participants were pre-screened and excluded from participation if they had a history of a significant neurological disorder, moderate to severe traumatic brain injury (TBI) or mild TBI in the past six months, lack of English proficiency, musculoskeletal impairments that would hinder performance on computer tasks, any current psychiatric disorders, or current alcohol or substance dependence. These criteria resulted in 30 DPs and 30 TD controls.

The web-based control group was acquired from the Richler *et al.* [23] dataset with the help of Dr Jeremy Wilmer. To be included in our analyses, we required that the controls be native English speakers between the ages of 22 and 70 years who completed the CFMT, NOMT Ziggerins and the face and object recognition questionnaire. These criteria resulted in 294 web controls. The age restriction was imposed to avoid possible age-related decline on the CFMT [31] and resulted in a sample with an age range identical to that of the DP group.

It is possible that some of the Richler *et al.* [23] web participants have DP, given that the prevalence rate is believed to be as high as 2.5% [32]. To avoid including potential DPs in our control sample, we removed any control participants who simultaneously scored *both* lower than 45 on the CFMT [31] *and* higher than 43 on the administered face recognition questionnaire (one standard deviation above the group mean, indicating worse self-reported face recognition; [23]), as well as anyone who had a reaction time greater than three standard deviations above the control mean on either task. Eleven web participants were removed for a combined low CFMT and high face recognition questionnaire, five were removed due to a high CFMT reaction time (greater than 7.48 s; 3 s.d. above mean), and four were removed due to a high NOMT reaction time (greater than 8.32 s; 3 s.d. above mean).[3] Based on these criteria, 274 control participants were included from the original online sample.

One notable testing difference between the web control sample and DPs is that the testing session was briefer in the web sample compared with DPs (20 min versus 3 h), with fatigue effects that may have influenced performance in DPs not applicable to the web sample [33]. While our in-lab controls performed the 3 h long battery in the same order as the DPs (with the CFMT administered first and the NOMT administered approximately halfway through), the web controls performed the NOMT and CFMT after only one short questionnaire. This suggests that our in-lab control group better accounts for potential fatigue or order effects than does the web-based control group.

Informed consent was obtained for all participants prior to data collection according to the Declaration of Helsinki. Participants were compensated for their time at a rate of $10 per hour. The study was approved by the VA Boston Healthcare System and Harvard Medical School Institutional Review Boards, and all study tasks were completed at either the VA Boston Healthcare System in Jamaica Plain or the Harvard Decision Science Lab.

## 2.3. Test battery

As our measure of face recognition ability, we used the original CFMT, a widely used and highly validated measure of face memory [17]. Using the CFMT as part of our diagnostic cut-off naturally created large group differences in CFMT performance between the DP group and both control groups ($p$s < 0.001). The benefit of using the CFMT as both a diagnostic measure and as our independent measure of face recognition ability is that we were able to create two groups with highly contrasting face recognition abilities, between which we can compare respective object recognition abilities. To measure object recognition abilities, we used a NOMT (Ziggerins; [23]). The NOMT was modelled after the CFMT and mirrors its structure by presenting the viewer with a novel object shown from three separate viewpoints, which the viewer must subsequently select from among three similar objects (figure 1). The trials repeat three consecutive times for each of the six target objects. After the initial 18 learning trials, all six learned objects are shown simultaneously for 20 s, followed by 30 test trials where the participant must select which from among three objects was one of the six they viewed previously. Finally, participants are shown another 20 s of the six learned objects followed by the remaining 24 test trials. For the final round of recognition trials, unlike the CFMT, the NOMT does not include visual noise intended to further obscure the stimuli.

---

[3]If the participants removed for potential DP are re-entered into the web sample ($N = 294$), the web control group significantly differs from the DPs' mean CFMT score ($M_{DP} = 38.30$, s.d._{DP} = 3.72, $M_{TD} = 57.27$ s.d._{TD} = 9.97, $p < 0.001$; $d = 2.52$), and, similar to the results reported below, does not significantly differ from the DPs' mean NOMT score ($M_{DP} = 58.20$, s.d._{DP} = 9.14, $M_{TD} = 61.07$, s.d._{TD} = 7.97, $p = 0.065$; $d = 0.33$).

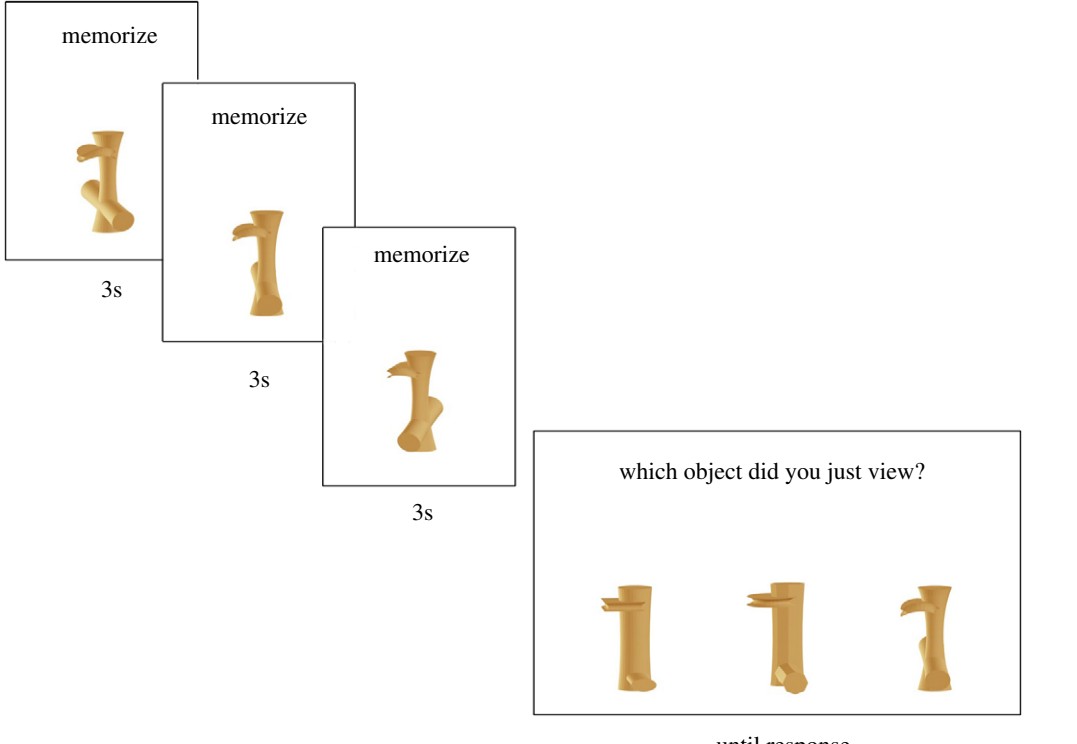

**Figure 1.** Learning phase for Novel Object Memory Test—Ziggerins. Adapted from Richler *et al.* [23], used with permission from Dr Isabel Gauthier.

All in-lab participants were tested on either a Lenovo laptop ($34.5 \times 19.5$ cm display, $1920 \times 1080$ pixels, 60 Hz) or a Sony Vaio ($34.29 \times 19.05$ cm, $1920 \times 1080$ pixels, 59 Hz).

## 2.4. Statistical approach

We performed independent samples *t*-tests between the DP and control groups for both CFMT and NOMT accuracy and reaction times to determine if DPs showed significant group-level differences on either task. To ensure a representative sample, we also compared the DPs' CFMT and NOMT scores and reaction times with a larger web-based control group [23]. We conducted additional Bayesian independent samples *t*-tests to better assess whether the results confirmed the null hypothesis. To check if the distribution of test scores and reaction times were equal between groups, we performed the Kolmogorov–Smirnov test of normality between the DP group and both control groups. Subsequently, we performed Mann–Whitney $U$ tests on the variables that showed unequal distribution across groups.

In addition to the group-level analyses, we also sought to determine whether DPs showed dissociations between tasks at the individual level. Using the Crawford *et al.* [24] stringent definition of a classical dissociation, an individual would need to (i) score significantly lower than controls on Task 1, (ii) show unimpaired performance on Task 2, and (iii) score significantly lower on Task 1 than on Task 2 ($p < 0.05$; two-tailed) to show a putative classical dissociation. If DPs show severely impaired performance on the CFMT (i.e. more than two standard deviations below the mean of the normative sample) while scoring normally on the NOMT (i.e. less than two standard deviations below the normative sample) with a significant difference between the two scores, then they could reasonably be classified as showing a dissociation between face and object recognition. We used the 30-person age- and gender-matched sample as our normative data. The in-lab sample was used in lieu of the web control sample due to its equivalence to the DPs in terms of the task battery and testing format. We also include results from the web sample in the supplementary materials. In addition to the dissociation analyses, we compared the prevalence of individual DP NOMT impairment with those of both the in-lab and web control groups using Fisher's exact test. To account for the difference in gender between the DP group and the web controls, we also divided the web

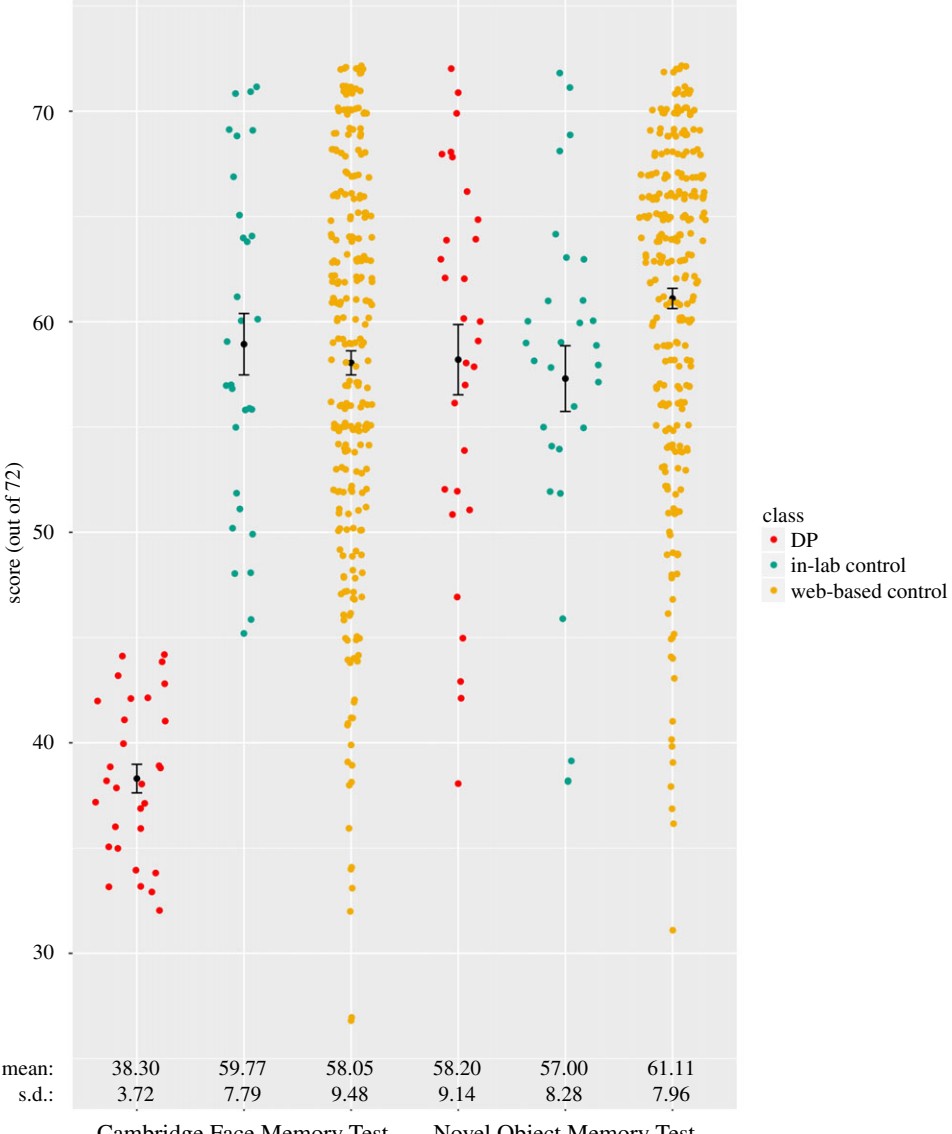

**Figure 2.** Individual raw scores of out 72 for DP, in-lab and web-based control participants performing the Cambridge Face Memory Test and Novel Object Memory Test.

controls by gender and compared individual DPs and controls with their respective gender on the NOMT (see electronic supplementary material).

# 3. Results

## 3.1. CFMT and NOMT group-level performance

First, we wanted to determine whether there were group-level differences on the CFMT and NOMT between the DPs and the matched in-lab controls. As expected, due to the qualification criteria, there was a significant difference between the two groups' CFMT scores ($t_{58} = -13.63$, $p < 0.001$; $d = 3.52$), with the DP mean score far below that of the in-lab controls ($M_{DP} = 38.30$, s.d.$_{DP} = 3.72$, $M_{TD} = 59.77$, s.d.$_{TD} = 7.79$). However, there were no differences between DP ($M_{DP} = 58.20$, s.d.$_{DP} = 9.14$) and in-lab control group means ($M_{TD} = 57.00$, s.d.$_{TD} = 8.28$) for the NOMT ($t_{58} = 0.53$, $p = 0.596$; $d = 0.14$) (figure 2). An additional Bayesian independent samples $t$-test revealed a Bayes factor of 4.52, indicating these results are 4.52 times more likely to be observed under the null hypothesis. CFMT reaction times were significantly slower for DPs than for in-lab controls ($t_{47} = 2.48$, $p = 0.017$; $d = 0.71$),

**Table 1.** Novel Object Memory Test and Cambridge Face Memory Test performance. Mean ± standard deviation. The *p*-values are derived from independent samples *t*-tests comparing DP and control test scores. **Statistically significant at $p < 0.05$.

| measure | DP | in-lab TD | web TD | *p*-values | |
|---|---|---|---|---|---|
| | | | | in-lab | web |
| CFMT score | 38.30 ± 3.72 | 59.77 ± 7.79 | 58.05 ± 9.48 | <0.001** | <0.001** |
| NOMT score | 58.20 ± 9.14 | 57.00 ± 8.28 | 61.11 ± 7.96 | 0.596 | 0.062 |
| CFMT reaction time (s) | 5.18 ± 1.56 | 4.13 ± 1.37 | 3.52 ± 1.03 | 0.017** | <0.001** |
| NOMT reaction time (s) | 4.21 ± 1.40 | 4.49 ± 1.49 | 4.07 ± 1.20 | 0.452 | 0.573 |

whereas NOMT reaction times were not significantly different between the two groups ($t_{58} = -0.76$, $p = 0.452$; $d = 0.20$) (table 1 and electronic supplementary material, figure S3). To determine whether DP and control groups were sampled from similar versus different distributions, we performed a Kolmogorov–Smirnov test for each variable. The Kolmogorov–Smirnov test of normality showed that the distribution of DP scores and reaction times differed from in-lab controls for CFMT accuracy ($D_{58} = 3.87$, $p < 0.001$), probably due to the constrained scores of the DP participants, but only showed a trend towards differing in CFMT reaction time ($D_{47} = 1.33$, $p = 0.059$), and did not differ between the groups for NOMT accuracy ($D_{58} = 0.90$, $p = 0.388$) or NOMT reaction time ($D_{58} = 0.78$, $p = 0.586$). A confirmatory Mann–Whitney $U$ test indicated that the difference between DP and in-lab control scores on the CFMT were significant for both accuracy ($U = 0.00$, $p < 0.001$) and reaction time ($U = 173.00$, $p = 0.017$) despite the variance in distribution.

We next compared the DP group results with the 274-person web sample. Similarly, we found that DPs as a group showed worse performance on the CFMT ($t_{302} = -22.24$, $p < 0.001$; $d = 2.74$), but only showed a trend towards a group-level difference on the NOMT ($t_{302} = -1.87$, $p = 0.062$; $d = 0.34$). A Bayesian independent samples *t*-test resulted in a Bayes factor of 1.31 in favour of the null hypothesis. Notably, the larger web sample differed from the DP group in terms of proportion of male versus female participants. When including gender as a covariate when comparing the NOMT accuracy between DP and web controls, the difference in scores failed to reach even a trend level of significance ($F_{1,297} = 2.56$, $p = 0.111$). A figure of CFMT and NOMT results as a function of gender and group is included in the supplementary materials (see electronic supplementary material, figure S1). NOMT reaction times for each group did not differ ($t_{302} = 0.51$, $p = 0.573$; $d = 0.10$), while DPs were significantly slower than controls for CFMT reaction times ($t_{292} = 4.68$, $p < 0.001$; $d = 1.26$) (table 1). A Kolmogorov–Smirnov test revealed that distributions of the DPs and web controls differed on CFMT accuracy ($D_{302} = 4.76$, $p < 0.001$) and reaction times ($D_{292} = 2.47$, $p < 0.001$) but not on NOMT accuracy ($D_{302} = 0.92$, $p = 0.371$) or reaction times ($D_{302} = 1.15$, $p = 0.141$). A Mann–Whitney $U$ test confirmed a significant difference between DP and web control scores on the CFMT ($U = 311.00$, $p < 0.001$).

## 3.2. CFMT and NOMT individual-level performance

We next sought to determine whether any DPs evinced classical dissociations between their face and novel object recognition [24]. Using the in-lab control group as our normative data, 22 out of 30 (73.3%) DPs showed a putative classical dissociation between their NOMT and CFMT scores. These individuals scored in the unimpaired range on the NOMT (*z*-score > −2) but were significantly impaired on the CFMT (*z*-score < −2), and their performance on the CFMT was significantly lower than their performance on the NOMT [34].

Out of the eight DPs who did not evince a classical dissociation, six had NOMT *z*-scores > −1.7, and seven had *z*-scores > −2. The remaining DP reached major impairment on the NOMT, scoring more than two standard deviations below the mean accuracy of the in-lab control group. Fisher's exact test shows this prevalence rate of 1 out of 30 DPs to be comparable with the 10% (3/30; $p = 0.612$) rate of NOMT impairment found in the in-lab control group, as well as the 2.6% (7/274; $p = 0.569$) prevalence found in the web control sample. If the threshold for deficit is lowered to more than 1.7 standard deviations below the mean (as used by Geskin & Behrmann [6]), DPs show a prevalence rate of 6.7% (2/30), failing to differ from either the in-lab controls (10%, 3/30; $p = 1.00$) or the web controls (2.9%, 8/274; $p = 0.258$) according to Fisher's exact test.

# 4. Discussion

The goal of the current study was to compare DPs' novel object recognition ability with that of controls using a validated and reliable assessment matched in format to the CFMT. At the group level, DPs' NOMT accuracy and reaction time did not significantly differ from either the in-lab or larger web-based control groups. Only 1 out of the 30 DPs had a NOMT $z$-score < −2 and 2 out of 30 DPs had $z$-scores < −1.7. These rates were not significantly different from the proportion of major and mild deficits found in either control group. Further, 22 out of the 30 DPs showed a classical dissociation between CFMT performance and NOMT performance, indicating severely impaired face recognition with preserved novel object recognition. Six of the remaining eight DPs had $z$-scores > −1.7 on the NOMT, suggesting that novel object recognition deficits are rare even in those not showing a classic dissociation. Together, these results observe a frequent dissociation across a large group of DPs, whereby face recognition impairment is associated with little to no impairment in a sensitive and well-validated test of novel object recognition that was designed to closely follow the structure of the CFMT. These results are consistent with those found with the 'blue objects' by Esins et al. [13], with the added benefits of a test more comparable to the CFMT and an absence of floor effects. Considering these findings with recent studies showing DP impairments in familiar object recognition tasks (e.g. cars), this suggests that DPs have normal general object perception and recognition abilities but that some DPs may have a decreased capacity to benefit from their experience with familiar object categories.

According to Gauthier, NOMTs measure general visual abilities that reflect the capacity to 'learn to match fingerprints (or) to perform radiological diagnosis' ([22], p. 72). Based on this, we interpret DPs' unimpaired performance on the NOMT Ziggerins as reflecting normal domain-general aspects of object perception and recognition abilities for a class of objects that, in controls, correlates highly with performance on other novel objects. Notably, the abilities required to successfully perform the NOMT do not depend on access to associative representations (e.g. names, semantic information) or rely on object representations learned over time, but instead involve object perception and memory processes that are unrelated to one's level of category-specific object experience [23]. DPs' normal performance on the NOMT Ziggerins probably reflects intact general abilities to perceive objects as well as to maintain object representations in memory over a delay. These intact mechanisms may at least partially underlie and support accurate perceptual and recognition performance with familiar object categories, particularly familiar objects associated with lower levels of expertise [23]. Consistent with this, a large study found DPs to be essentially normal during sequential perceptual matching of familiar objects including hands and houses (while being impaired at face matching, [4]) and a smaller study found no difference between DPs' and controls' accuracy in the simultaneous matching of both self and others' hands and feet [35]. Further, larger studies of DPs have found that DP and control groups did not differ on familiar object recognition performance for bicycles (Cambridge Bicycle Memory Test; [10]), shoes [12], shells [13] or flowers [14].

DPs' normal group-level NOMT Ziggerins performance and previous reports of normal familiar object perception and recognition abilities (e.g. [4]) are also consistent with neural results comparing DPs and controls. When examining resting-state functional MRI connectivity in the visual 'object network', Song and colleagues found no differences between DPs and controls, though they found decreased connectivity among core face-selective regions [14]. Further, when analysing an fMRI localizer task using short movie clips of faces, objects and bodies, Jiahui et al. [36] found normal familiar object selectivity (familiar objects versus faces) in ventral temporal cortex in DPs, though interestingly they found reduced body selectivity in DPs (bodies versus familiar objects) as well as reduced face selectivity [36]. Event-related potential studies have also shown normal DP responses to objects in the N170 (e.g. houses: [37]), though not to bodies [38]. Together, these neural findings are consistent with the current NOMT results and provide converging evidence that DPs have intact general object processing abilities that support the perception and recognition of familiar object categories with lower levels of expertise.

Despite evidence for normal object perception and recognition mechanisms in DPs as indexed by the NOMT, a significant number of studies have demonstrated impaired DP performance for certain familiar object categories, suggesting that DPs' object perception and recognition may not benefit as much from experience [10]. This may particularly affect familiar objects with very high levels of experience, such as bodies and cars [18]. For example, a study by Biotti et al. [11] found that DPs have significant group-level deficits when perceptually matching bodies and cars. Further, Rivolta et al. [39] showed that while DPs' body perception accuracy did not differ from controls', their reaction times were significantly slower [39]. With regard to object recognition, a large study found that DPs as a group had reduced car recognition

performance [15] and a similar finding was observed in smaller studies [19,20]. Additionally, using a long-term recognition task with cars, Barton *et al.* [10] found that DPs had reduced performance compared with controls when taking into account DPs' car expertise. Specifically, for every increment in car expertise as indexed by verbal semantic knowledge, DPs had only about half the gains in car recognition accuracy that were shown by healthy controls. Because object category expertise is rarely measured in DP studies, it is difficult to know if DPs' deficits are specific to object categories with very high levels of expertise such as cars and bodies or rather are more common across familiar object categories, perhaps explaining heterogeneous performance in less familiar object categories. It would be useful for future studies to examine object perception and recognition using categories ranging from less familiar to highly familiar while assessing DPs' and controls' category expertise (similar to [10]).

DPs' normal NOMT performance existing alongside deficits in expertise-dependent familiar object perception and recognition could have several explanations. First, studies have linked greater holistic perceptual processing to better recognition ability (e.g. [40]) and familiar objects such as cars and bodies have shown to be processed in a more holistic manner than more novel objects, showing significant effects of inversion (cars: [41]; bodies: [42]; Greebles in Greeble 'experts': [43]). Interestingly, DPs have shown to have similar, albeit modest inversion effects with cars compared with controls [44], but evidence of disrupted N170 inversion effects with bodies [38], suggesting potential deficits in holistic processing of familiar objects. These holistic processing deficits may contribute to DPs' poorer recognition memory for familiar objects by making familiar objects less perceptually distinct. An alternative explanation is that DPs are less able to make semantic or contextual associations with highly familiar objects. These associations could help with more elaborative encoding of familiar objects and better subsequent recognition [45]. This interpretation is consistent with the conceptualization of DP as a disconnection syndrome between perceptual regions in the ventral temporal cortex and more anterior temporal regions involved in storing semantic knowledge [46].

The presence of deficits in some DPs for highly familiar objects while exhibiting normal NOMT performance has an interesting resemblance to the case of AW, the only reported case of selective developmental object agnosia (DOA; [47]). AW showed severe impairments in recognition memory across numerous object categories, including doors, cars, guns and tools, though not all categories, houses and sunglasses were unimpaired. Notably, AW was able to perform normally at a novel object recognition test, similar to DPs in the current study. AW's normal performance on novel object recognition while being impaired in recognizing certain categories of familiar objects parallels findings in DPs and suggests that DOA and DP (or at least some cases of DP) may involve similar mechanisms. This could be because these disorders co-occur or rather because they may disrupt overlapping mechanisms along the ventral visual/anterior temporal lobe processing stream (see [48], for a discussion). To better understand potential similarities between DOA and DP, it would be useful for future studies to compare object perception and recognition of a broad array of familiar object categories (while taking into account object expertise) as well as assess performance using identical NOMTs.

It should be noted that though DPs have shown group-level deficits in perceiving and recognizing familiar objects, such as with cars and bodies (e.g. [10,15]), the proportion of individual DPs showing impairments are often only a small minority (z-scores < −2: 4.3%; [15]; 16.7%; [10]; 22.0%, [6]). Thus, the majority of DPs have normal familiar object perception and recognition abilities, even with highly familiar objects. Additionally, caution should be used when interpreting an individual's performance on any single familiar object category (as is often done in case analyses in DP studies), since even performance in unimpaired participants may be heterogeneous across familiar object categories, probably due to differences in experience and expertise. For example, Edward, a severe DP, was able to perform normally on multiple old/new tasks of familiar object recognition including horses, tools and houses, and was able to successfully gain expertise with novel objects in a Greebles training programme [5]. However, he performed poorly on an old/new task with guns. Had Edward been only assessed on gun performance, his normal object perception, learning and recognition abilities apart from guns would have been missed. Additionally, it is notable that even in the DOA AW, there were familiar object categories, including houses and sunglasses, for which her scores were completely normal. Thus, future studies with large samples of DPs employing perceptual and recognition tasks with a broad range of familiar object categories would be useful to determine the prevalence and heterogeneity of DPs' familiar object recognition difficulties.

Though the findings of the current study are compelling and offer a novel perspective on DPs' object processing abilities, this study has several limitations. First, we used only a single novel object recognition task. Although NOMTs may be more representative of category-general processes, it

would have been better to test DPs with multiple NOMTs to better quantify this ability. While NOMTs have shown to correlate highly with one another [23] and previous case studies have found normal DP performance with Greebles [49], further investigations would be useful to confirm the generalizability of normal NOMT performance in DPs. Second, the current study only assessed novel object recognition and it would be important for future studies to test both NOMTs and several categories of familiar object recognition as well as familiar object perceptual matching tasks (while factoring in category expertise) to more fully elucidate DPs' object processing abilities.

In sum, we show that as a group DPs perform equally as well as control subjects on a test of novel object memory, and that the proportion of impaired DPs was not significantly different from controls. This finding, combined with recent DP studies of familiar object recognition, suggests that object recognition deficits do not inevitably accompany face recognition impairments. Instead, they suggest that DPs' previously reported familiar object perception and recognition deficits stem from a decreased perceptual or mnemonic capacity to benefit from their experience with familiar object categories.

Data accessibility. Supporting data is available in the Dryad Digital Repository at: https://doi.org/10.5061/dryad. 59zw3r25d [52].

Authors' Contributions. R.F. carried out data collection, participated in data analysis and drafted the manuscript; J.W. provided additional data, designed and provided figures and critically revised the manuscript; I.X. carried out data collection and assisted in drafting the manuscript; M.V. critically revised the manuscript; J.D. conceived of the study, designed and coordinated the study, and helped draft the manuscript.

Competing Interests. The authors report no competing interests.

Funding. This study was funded by R01 from the National Eye Institute grant no. RO1EY026057 awarded to J.D. M.V.'s effort was supported by a Senior Research Career Scientist award from the Clinical Science Research and Development Service, Department of Veterans Affairs. The contents of this manuscript do not represent the views of the US Department of Veterans Affairs or the United States Government.

Acknowledgements. We want to thank our developmental prosopagnosic and control participants for completing our challenging battery of tasks.

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
