## [Reviewer comments · Royal Society Open Science]

Review History

RSOS-200988.R0 (Original submission)

Review form: Reviewer 1

Is the manuscript scientifically sound in its present form?

Yes

Are the interpretations and conclusions justified by the results?

Yes

Is the language acceptable?

Yes

Do you have any ethical concerns with this paper?

No

Have you any concerns about statistical analyses in this paper?

Yes

Recommendation?

Accept with minor revision (please list in comments)

Comments to the Author(s)

The manuscript investigates the face-selectivity of recognition difficulties in DP, by contrasting face abilities on the CFMT with performance on a (mostly) matched measure of recognition for novel objects: to limit any moderating influence of previous experience. The paper was very well written. I liked how the authors compared the DP group with a group of matched controls (tested in the lab) as well as a larger and more representative normative sample (tested online as part of a previous study), and also how they conducted both group- and individual-level analyses to probe patterns of performance. Overall, I thought the study was well-motivated, the results were interesting and the conclusions seemed appropriate. I have only a few, fairly minor comments/suggestions.

Re. the in-lab participant details. I might have missed something, but how was participant education measured exactly? Also, I found myself looking for the details of the matching (i.e., statistical tests) in the participants section, rather than results. It is obviously only a minor point, but if this were reported earlier, you could avoid the repetition of reporting mean ages etc.

The details of how many additional participants (DP and lab-controls) were excluded from participating based on the experimenters' various criteria could helpfully be made explicit.

Did you happen to investigate the pattern of performance on the CFMT and NOMT in those participants in the web-based group that were excluded due to concerns about them being potentially DP? Obviously, they weren't recruited using the same stringent criteria as the main clinical sample, but I wonder whether they still might be interesting to report somewhere? Even as a footnote, in the individual level analysis perhaps?

It would be good to include effect size estimates for the t-tests etc on pages 10 - 13.

Where Kolmogorov-Smirnov tests of normality identify a significant difference in the distribution of DP and in-lab control scores, then is it appropriate to report confirmatory non-parametric tests?

Typo - page 6 "vocabularyly"

Review form: Reviewer 2

Is the manuscript scientifically sound in its present form?

No

Are the interpretations and conclusions justified by the results?

No

Is the language acceptable?

Yes

Do you have any ethical concerns with this paper?

No

Have you any concerns about statistical analyses in this paper?

No

Recommendation?

Major revision is needed (please make suggestions in comments)

Comments to the Author(s)

The paper addresses an interesting and topical question- whether individuals with developmental prosopagnosia show object recognition deficits, as well as face recognition deficits. The authors use a novel object recognition test in a large sample of DPs. No differences in performance were found between DPs and two control samples, suggesting normal novel object recognition in DP. Despite this being a topical question, I believe this paper makes a novel contribution to the literature.

I have some questions over the structure of the report. It is typical to have the participants described (including differences between groups on diagnostics) in the participant section, rather than the results. This would work well for the current report, and mean that all demographic data could be covered for the three groups in one go (it's unclear from the text how many males/females there are in the lab control sample vs. the DP sample at the moment). I strongly believe that the CFMT group analyses should be in the methods section - we know there is a difference between the groups - they have been artificially manufactured this way. The CFMT group difference should be removed from the results and placed in the methods under the participants section where the samples should be described in full. The current structure is misleading, as it implies the CFMT is a dependent variable, where it is actually the means by which the samples are grouped. The abstract would also need to reflect this change.

It would be very helpful to have Bayesian analyses on the between group t-tests for the NOMT task. This would provide the strength of evidence in favour of the null hypothesis for the critical null t-tests.

Supplementary Figure 2 or 3 should be shown in the paper, so readers can easily inspect the novel objects. If the data are based only on the Ziggerins, only these objects should be presented.

Why did you choose to use Ziggerins? I just wonder whether there might be something specific about their visual appearance (the wooden joints) that make them easier for DPs to process than other novel objects. This isn't a problem, as it's interesting to know how far the impairments in DP extend, but it would be helpful to have some discussion on the visual properties of these objects. Relatedly, in the discussion, the authors generalise too much from the Ziggerins they've tested. For example: Page 13, line 53, You can't say with any certainty that because a high correlation has previously been found between the tasks, DPs will be unimpaired on all. Page 14, line 17, Was this in reference to all NOMTs? Are you sure this holds for only Ziggerins?

Minor comments:

Page 8, line 3: add some details on the questionnaire that was used; where was it previously published?

Use a consistent number of decimal places throughout.

It's interesting that even given the demands for the web-based task were much lower than the in-person testing, many of the web-controls performed very poorly on the CFMT. Do the authors have any thoughts on this?

Page 13, line 45. Begin sentence with 'Six'

Decision letter (RSOS-200988.R0)

Dear Ms Fry

On behalf of the Editors, I am pleased to inform you that your Manuscript RSOS-200988 entitled "Evidence for normal novel object recognition abilities in developmental prosopagnosia" has been accepted for publication in Royal Society Open Science subject to minor revision in accordance with the referee suggestions. Please find the referees' comments at the end of this email.

The reviewers and handling editors have recommended publication, but also suggest some minor revisions to your manuscript. Therefore, I invite you to respond to the comments and revise your manuscript.

- Ethics statement

- Data accessibility

<http://datadryad.org/submit?journalID=RSOS&manu=RSOS-200988>

- Competing interests

- Authors' contributions

AB carried out the molecular lab work, participated in data analysis, carried out sequence alignments, participated in the design of the study and drafted the manuscript; CD carried out

the statistical analyses; EF collected field data; GH conceived of the study, designed the study, coordinated the study and helped draft the manuscript. All authors gave final approval for publication.

- Acknowledgements

- Funding statement

Because the schedule for publication is very tight, it is a condition of publication that you submit the revised version of your manuscript before 29-Jul-2020. Please note that the revision deadline will expire at 00.00am on this date. If you do not think you will be able to meet this date please let me know immediately.

If your manuscript is newly submitted and subsequently accepted for publication, you will be asked to pay the article processing charge, unless you request a waiver and this is approved by Royal Society Publishing. You can find out more about the charges at <https://royalsocietypublishing.org/rsos/charges>. Should you have any queries, please contact openscience@royalsociety.org.

on behalf of Dr Teodora Gliga (Associate Editor) and Essi Viding (Subject Editor)
openscience@royalsociety.org

Associate Editor Comments to Author (Dr Teodora Gliga):

Comments to the Author:

I have now received reports from two experts in the fields. They concur in finding your paper rigorous and novel. Both provide recommendations for how to improve the organisation of the paper and the sample and statistical reporting (including Bayesian stats and reporting effect sizes), which I expect you to address. Please provide a point by point response to all reviewer's comments.

Reviewer comments to Author:

Reviewer: 1

Comments to the Author(s)

The manuscript investigates the face-selectivity of recognition difficulties in DP, by contrasting face abilities on the CFMT with performance on a (mostly) matched measure of recognition for novel objects: to limit any moderating influence of previous experience. The paper was very well written. I liked how the authors compared the DP group with a group of matched controls (tested in the lab) as well as a larger and more representative normative sample (tested online as part of a previous study), and also how they conducted both group- and individual-level analyses to

probe patterns of performance. Overall, I thought the study was well-motivated, the results were interesting and the conclusions seemed appropriate. I have only a few, fairly minor comments/suggestions.

Re. the in-lab participant details. I might have missed something, but how was participant education measured exactly? Also, I found myself looking for the details of the matching (i.e., statistical tests) in the participants section, rather than results. It is obviously only a minor point, but if this were reported earlier, you could avoid the repetition of reporting mean ages etc.

The details of how many additional participants (DP and lab-controls) were excluded from participating based on the experimenters' various criteria could helpfully be made explicit.

Did you happen to investigate the pattern of performance on the CFMT and NOMT in those participants in the web-based group that were excluded due to concerns about them being potentially DP? Obviously, they weren't recruited using the same stringent criteria as the main clinical sample, but I wonder whether they still might be interesting to report somewhere? Even as a footnote, in the individual level analysis perhaps?

It would be good to include effect size estimates for the t-tests etc on pages 10 – 13.

Where Kolmogorov-Smirnov tests of normality identify a significant difference in the distribution of DP and in-lab control scores, then is it appropriate to report confirmatory non-parametric tests?

Typo – page 6 “vocabularyly”

Reviewer: 2

Comments to the Author(s)

The paper addresses an interesting and topical question- whether individuals with developmental prosopagnosia show object recognition deficits, as well as face recognition deficits. The authors use a novel object recognition test in a large sample of DPs. No differences in performance were found between DPs and two control samples, suggesting normal novel object recognition in DP. Despite this being a topical question, I believe this paper makes a novel contribution to the literature.

I have some questions over the structure of the report. It is typical to have the participants described (including differences between groups on diagnostics) in the participant section, rather than the results. This would work well for the current report, and mean that all demographic data could be covered for the three groups in one go (it's unclear from the text how many males/females there are in the lab control sample vs. the DP sample at the moment). I strongly believe that the CFMT group analyses should be in the methods section – we know there is a difference between the groups – they have been artificially manufactured this way. The CFMT group difference should be removed from the results and placed in the methods under the participants section where the samples should be described in full. The current structure is misleading, as it implies the CFMT is a dependent variable, where it is actually the means by which the samples are grouped. The abstract would also need to reflect this change.

It would be very helpful to have Bayesian analyses on the between group t-tests for the NOMT task. This would provide the strength of evidence in favour of the null hypothesis for the critical null t-tests.

Supplementary Figure 2 or 3 should be shown in the paper, so readers can easily inspect the novel objects. If the data are based only on the Ziggerins, only these objects should be presented.

Why did you choose to use Ziggerins? I just wonder whether there might be something specific about their visual appearance (the wooden joints) that make them easier for DPs to process than other novel objects. This isn't a problem, as it's interesting to know how far the impairments in DP extend, but it would be helpful to have some discussion on the visual properties of these objects. Relatedly, in the discussion, the authors generalise too much from the Ziggerins they've tested. For example: Page 13, line 53, You can't say with any certainty that because a high correlation has previously been found between the tasks, DPs will be unimpaired on all. Page 14, line 17, Was this in reference to all NOMTs? Are you sure this holds for only Ziggerins?

Minor comments:

Page 8, line 3: add some details on the questionnaire that was used; where was it previously published?

Use a consistent number of decimal places throughout.

It's interesting that even given the demands for the web-based task were much lower than the in-person testing, many of the web-controls performed very poorly on the CFMT. Do the authors have any thoughts on this?

Page 13, line 45. Begin sentence with 'Six'

Author's Response to Decision Letter for (RSOS-200988.R0)

See Appendix A.

Decision letter (RSOS-200988.R1)

Dear Ms Fry:

On behalf of the Editors, I am pleased to inform you that your Manuscript RSOS-200988.R1 entitled "Evidence for normal novel object recognition abilities in developmental prosopagnosia" has been accepted for publication in Royal Society Open Science subject to minor revision in accordance with the referee suggestions. Please find the referees' comments at the end of this email.

The reviewers and Subject Editor have recommended publication, but also suggest some minor revisions to your manuscript. Therefore, I invite you to respond to the comments and revise your manuscript.

- Ethics statement

- Data accessibility

<http://datadryad.org/submit?journalID=RSOS&manu=RSOS-200988.R1>

- Competing interests

- Authors' contributions

- Acknowledgements

- Funding statement

Because the schedule for publication is very tight, it is a condition of publication that you submit the revised version of your manuscript before 12-Aug-2020. Please note that the revision deadline will expire at 00.00am on this date. If you do not think you will be able to meet this date please let me know immediately.

To revise your manuscript, log into <https://mc.manuscriptcentral.com/rsos> and enter your Author Centre, where you will find your manuscript title listed under "Manuscripts with

Decisions". Under "Actions," click on "Create a Revision." You will be unable to make your revisions on the originally submitted version of the manuscript. Instead, revise your manuscript and upload a new version through your Author Centre.

- 1) A text file of the manuscript (tex, txt, rtf, docx or doc), references, tables (including captions) and figure captions. Do not upload a PDF as your "Main Document".
- 2) A separate electronic file of each figure (EPS or print-quality PDF preferred (either format should be produced directly from original creation package), or original software format)
- 3) Included a 100 word media summary of your paper when requested at submission. Please ensure you have entered correct contact details (email, institution and telephone) in your user account
- 4) Included the raw data to support the claims made in your paper. You can either include your data as electronic supplementary material or upload to a repository and include the relevant doi within your manuscript
- 5) All supplementary materials accompanying an accepted article will be treated as in their final form. Note that the Royal Society will neither edit nor typeset supplementary material and it will be hosted as provided. Please ensure that the supplementary material includes the paper details where possible (authors, article title, journal name).
- 6) Access to your de-identified data needs to be provided upon re-submission. We cannot proceed with your manuscript submission until this data has been provided.

on behalf of Dr Teodora Gliga (Associate Editor) and Essi Viding (Subject Editor)
openscience@royalsociety.org

Associate Editor Comments to Author (Dr Teodora Gliga):

Comments to the Author:

Thank you for addressing the reviewers' comments and for submitting to our journal.

Author's Response to Decision Letter for (RSOS-200988.R1)

See Appendix B.

Decision letter (RSOS-200988.R2)

Dear Ms Fry,

It is a pleasure to accept your manuscript entitled "Evidence for normal novel object recognition abilities in developmental prosopagnosia" in its current form for publication in Royal Society Open Science. The comments of the reviewer(s) who reviewed your manuscript are included at the foot of this letter.

on behalf of Dr Teodora Gliga (Associate Editor) and Essi Viding (Subject Editor)
openscience@royalsociety.org

Appendix A

Evidence for normal novel object recognition abilities in developmental
prosopagnosia

Regan Fry^{1,2}, Jeremy Wilmer³, Isabella Xie^{4,5}, Mieke Verfaellie^{6,7}, Joseph DeGutis^{1,2}

1. Boston Attention and Learning Laboratory, VA Boston Healthcare System, Boston, Massachusetts, United States of America
2. Department of Psychiatry, Harvard Medical School, Boston, Massachusetts, United States of America
3. Wellesley College, Wellesley, Massachusetts, United States of America
4. Washington University in St. Louis, St. Louis, Missouri, United States of America
5. Harvard Decision Science Lab, Harvard Kennedy School, Cambridge, Massachusetts, United States of America
6. Memory Disorders Research Center, VA Boston Healthcare System, Boston, Massachusetts, United States of America
7. Boston University School of Medicine, Department of Psychiatry, Boston, Massachusetts, United States of America

Corresponding Author:

Regan Fry

Regan_Fry@hms.harvard.edu

150 S. Huntington Ave., 182JP

Boston, MA 02130

Abstract

The issue of the face specificity of recognition deficits in developmental prosopagnosia (DP) is fundamental to the organisation of high-level visual memory and has been increasingly debated in recent years. Previous DP investigations have found some evidence of object recognition impairments, but have almost exclusively used familiar objects (e.g., cars), where performance may depend on acquired object-specific experience and related visual expertise. An object recognition test not influenced by experience could provide a better, less contaminated measure of DPs' object recognition abilities. To investigate this, in the current study we tested 30 DPs and 30 matched controls on a novel object memory test (NOMT Ziggerins) and the Cambridge Face Memory Test (CFMT). DPs ~~with severe~~ ~~were impair~~ ~~impaired~~ on the CFMT ~~but~~ showed no differences in accuracy or reaction times to controls on the NOMT. We found similar results when comparing DPs to a larger sample of 274 web-based controls. Additional individual analyses demonstrated that the rates of object recognition impairment in DPs did not differ from the rate of impairment in either control group. Together, these results demonstrate unimpaired object recognition in DPs for a class of novel objects that serves as a powerful index for broader novel object recognition capacity.

Keywords: developmental prosopagnosia, object recognition, face specificity, novel object memory test

Abbreviations: DP = developmental prosopagnosia; NOMT = novel object memory test, CFMT = Cambridge Face Memory Test

Introduction

Developmental prosopagnosia (DP) is a neurodevelopmental disorder resulting in lifelong face recognition deficits in the absence of brain injury or co-occurring social, intellectual, or visual impairments (Behrmann & Avidan, 2005; Duchaine & Nakayama, 2006b; Susilo & Duchaine, 2013). The controversy over whether DP is specific to faces or involves additional domain-general deficiencies has been raised a number of times in the last 20 years (Bate, Bennetts, Tree, Adams, & Murray, 2019; Duchaine, Yovel, Butterworth, & Nakayama, 2006; Geskin & Behrmann, 2017), particularly regarding the presence of co-occurring deficits in object recognition.¹ Evidence has shown that at least some DPs have highly face-specific deficits, such as the case of Edward, a DP with preserved familiar object recognition and recall despite profound impairments in face perception and recognition (Duchaine et al., 2006; for others, see Wegrzyn, Garlichs, Heß, Woermann, & Labudda, 2019; Bentin, Deouell, & Soroker, 1999). These DP cases clearly show that face recognition can be dissociated from familiar object recognition, providing important evidence that they rely on distinct mechanisms.

Despite these clear cases of dissociation, studies suggest that face and familiar object recognition deficits in DPs may co-occur. DPs have shown higher familiar object recognition impairment rates than controls at the individual level (Geskin & Behrmann, 2017). In a comprehensive meta-analysis of DP cases, Geskin and Behrmann examined a wide range of familiar object categories and found that, of those studies with both accuracy and reaction time measures, 80.3% (191/238) of DPs experienced co-occurring object recognition deficits (either accuracy *or* reaction time z -score < -1.7). When including all DPs with accuracy data², excluding DPs with only a reaction time impairment, and using a more standard $z < -2$ cutoff, the DP familiar object recognition impairment rate was still a substantial 22.0% (101/459), much higher than the 2.5% rate predicted in controls.

In studies of familiar object recognition, DPs have often shown reduced, though normal group-level performance for object categories such as bicycles (Barton, Albonico, Susilo,

¹ The term “recognition” has sometimes been used to refer to perceptual matching (e.g., the Benton Face Recognition Test) and other times to memory (e.g., Warrington Recognition Memory Test for Faces). In this paper, we use “recognition” as it refers to memory, specifically “the ability to identify information as having been encountered before” (APA Dictionary of Psychology).

² We recalculated impairment based on accuracy alone because nearly all of these object recognition tests were given without instructions to perform as quickly as possible and also because it has yet to be shown that reaction time on any of these tests explains unique variance beyond accuracy in DP diagnosis or CFMT scores.

Duchaine, & Corrow, 2019; Biotti, Gray, & Cook, 2017), shoes (Stollhoff, Jost, Elze, & Kennerknecht, 2011), shells (Esins, Schultz, Wallraven, & Bülhoff, 2014), and flowers (Song, Zhu, Li, Wang, & Liu, 2015). However, in a recent large study of 46 DPs, Gray and colleagues (2019) found that, compared to controls, the DP group scored significantly worse (Cohen's $d = .49$) on the Cambridge Car Memory Test (CCMT; Dennett, McKone, Tavashmi, Hall, Pidcock, Edwards, & Duchaine, 2012; Gray, Biotti, & Cook, 2019). The CCMT is a sensitive and widely used test of familiar object recognition that matches the Cambridge Face Memory Test (CFMT; Duchaine & Nakayama, 2006a) in format, and cars are a category of objects for which participants typically have very high levels of experience and expertise (McGugin, Richler, Herzmann, Speegle, & Gauthier, 2012). Similar-sized decrements in car recognition have also been observed in smaller samples of DPs (e.g., Gerlach, Klargaard, & Starrfelt, 2016; Tanzer, Weinbach, Mardo, Henik, & Avidan, 2016; though not all, see Esins, Schulz, Stemper, Kennerknecht, & Bulthoff, 2016). Interestingly, Barton et al. (2019) found that DPs and controls had similar car recognition performance on a long-term memory task assessing recognition of familiar cars, but when taking into account semantic car knowledge, DPs showed significantly reduced performance compared to controls. Although the long-term recognition task from Barton et al. (2019) differs from shorter-term episodic memory tasks typically used to assess familiar object recognition, these results raise the interesting possibility that DPs' familiar object recognition impairments, when observed, may be due to a failure to benefit from experience with these objects rather than to a domain-general impairment in object recognition.

To test the hypothesis that DPs have an impairment in experience-dependent learning of familiar objects, we used a category of objects for which no-one has experience: novel objects. Novel objects are laboratory-created stimuli that have similar overall shape and can be discriminated based on parts that vary between individual cases (e.g., rounded vs. rectangular projections, double- vs. single-sided edges). Because novel objects are not influenced by category-specific experience or expertise, they are thought to better measure the domain-general visual capacity to learn and recognise objects across all categories (Gauthier, 2018). Supporting this, Richler, Wilmer, and Gauthier (2017) demonstrated that three versions of a novel object memory test (NOMT) all show more shared domain-general variance (average 23%) than tests using familiar objects (e.g., cars, horses, guns; average 10% shared variance). NOMTs are uniquely interesting in that one's performance is thought to reflect visual perception, learning,

and memory abilities without the moderating effects of prior experience (Richler et al., 2017; Gauthier, 2018). Performing NOMTs with DPs could have important theoretical importance; if DPs show worse group-level performance than controls on NOMTs, this would be indicative of more general deficits in perception, encoding, or retrieval of objects. Conversely, normal group-level performance would suggest that DPs' category-general object abilities are intact and that impairments in recognition of highly familiar objects (e.g., cars) are more likely due to an inability to benefit from experience. One previous study tested DPs and controls using a novel object recognition memory test ('blue objects', Esins, Schults, Wallraven, & Bulthoff, 2014). As a group, DPs performed comparably to controls but mean accuracy was very low in both groups ($d' = .6$), suggestive of floor effects. This makes it challenging to rule out differences between DPs and controls. An aim of the current study was to better assess DPs vs. control differences by using a novel object recognition memory test with a broad range of scores and better overall performance.

In the current study, we sought to characterise DP's novel object recognition abilities by administering the NOMT Ziggerins and the similarly formatted CFMT to a sample of 30 DPs and 30 matched and matched controls. The Ziggerins were chosen because they bear no resemblance to either faces or familiar objects, have high internal reliability (Cronbach's $\alpha = 0.89$; Richler et al., 2017), and are free from floor effects. Performance on Ziggerins also correlates highly with other measures of visual object recognition while dissociating from vocabulary and digit span, suggesting that general verbal ability does not significantly contribute to Ziggerins performance (Richler et al., 2017). Further, we compared DP scores to 274 controls from a recent study by Richler and colleagues (Richler et al., 2017). In addition to these group-level analyses, we employed case-based analyses to determine whether (1) individual DPs showed impairment on the NOMT Ziggerins, and (2) any DPs showed a classical dissociation between object and face recognition (Crawford, Garthwaite, & Gray, 2003). By comparing the performance of DPs and typically developed (TD) controls on the CFMT and a NOMT at both the group and individual levels, we sought to characterise novel object recognition in DPs.

Methods

Participants:

Running Head: OBJECT RECOGNITION PROSOPAGNOSIA

Participants were between the ages of 18 and 70 years old ($N=60$). ($N=60$, $N_{\text{Female}}=43$; $M_{\text{DP}}=38.5$, $SD_{\text{DP}}=13.69$, $M_{\text{TD}}=40.03$, $SD_{\text{TD}}=11.73$). Developmental prosopagnosics were recruited from our database of previous DP participants in the Boston area, references from other research labs (Dr. Matthew Peterson, MIT; Dr. Brad Duchaine, Dartmouth College, www.faceblind.org), and individuals who responded to our advertisement on the Massachusetts Bay Transportation Authority subway system. Control subjects were recruited from both the Harvard Decision Science Laboratory in Cambridge, Massachusetts and through flyers distributed in the Boston area.

The DP group did not differ from the in-lab control group in either age ($M_{\text{DP}}=38.50$, $SD_{\text{DP}}=13.69$, $M_{\text{TD}}=40.03$, $SD_{\text{TD}}=11.73$, $p=.643$), gender ($t(57) = 1.70$, $p = .095$), or education ($M_{\text{DP}}=17.52$, $SD_{\text{DP}}=2.57$, $M_{\text{TD}}=16.17$, $SD_{\text{TD}}=2.80$; $t(57) = 1.93$, $p = .059$). Education was measured for the in-lab groups by years of formal education, with options ranging from 1st grade (coded as 1) to 4th year of graduate school (coded as 20). The DP group did not differ from the web controls in in age ($M_{\text{Web}} = 36.78$, $SD_{\text{Web}} = 12.04$, $p = .464$), however, the DP group had a higher proportion of females ($N_{\text{Female}} = 24$, 80.0%) than the web control group ($N_{\text{Female}} = 160$, 58.2%; $t(298) = 3.06$, $p = .004$). Results separated by gender can be found in the supplementary materials (see Supplementary Figure 1). The web controls also had a lower average education level ($M_{\text{DP}}=4.52$, $SD_{\text{DP}}=.95$, $M_{\text{TD}}=3.76$, $SD_{\text{TD}}=1.04$; $t(273) = 3.74$, $p < .001$). The web controls' education was measured on a scale of 1 to 5, where 1 = Middle school, 2 = High school/Secondary school, 3 = Some college/University, 4 = Bachelor's degree, and 5 = Graduate degree. For comparison between the two groups, DP education levels were re-coded to match this scale. While ideally an education-matched web control group is preferred, studies have shown that face recognition is independent from both education and IQ (Wilmer, 2017). Similarly, education does not predict any additional variance in general memory performance outside of age (West, Crook, & Barron, 1992).

DP and Control Qualifications:

Developmental prosopagnosics were screened using the 20-Item Prosopagnosia Index (PI-20; Shah, Gaule, Sowden, Bird, & Cook, 2015), a famous faces memory test (FFMT), and the CFMT (Duchaine & Nakayama, 2006a). To qualify as a DP, participants had to have scored above 64 on the PI-20, more than two standard deviations below the control mean on the Famous Faces Memory Test, and 44 or lower out of 72 on the original CFMT (Garrido, Duchaine, &

Formatted: Indent: First line: 3 pi

Formatted: Font: Italic

Formatted: Font color: Auto

Nakayama, 2008; Duchaine, Yovel, & Nakayama, 2007). Seven potential DPs were excluded on the basis of CFMT score. The PI-20 and FFMT were given as a pre-screening measure, while and the CFMT was included as part of the battery. All participants had normal or corrected-to-normal vision, and had to have scored within the normal range on the Autism Spectrum Quotient (< 33; Baron-Cohen, Wheelwright, Skinner, Martin, & Clubley, 2001) and the Leuven Perceptual Organization Screening Test (L-POST; Torfs, Vancleef, Lafosse, Wagemans, & de-Wit, 2014) to rule out other causes of poor face recognition. No DPs were excluded on the basis of L-POST score. In-lab controls had similar criteria with the exception that they must *not* report any lifelong face recognition issues, score > 64 on the PI-20, or score lower than 45 on the CFMT. Control subjects were only invited to participate in the entire study if they met the above criteria. Participants were pre-screened and excluded from participation if they had a history of a significant neurological disorder, moderate to severe traumatic brain injury (TBI) or mild TBI in the past 6 months, lack of English proficiency, musculoskeletal impairments that would hinder performance on computer tasks, any current psychiatric disorders, or current alcohol or substance dependence. These criteria resulted in 30 DPs and 30 typically developed controls.

The web-based control group was acquired from Richler and colleagues (2017) dataset with the help of Dr. Jeremy Wilmer. To be included in our analyses, we required that the controls be native English speakers between the ages of 22 and 70 years who completed the CFMT, NOMT Ziggerins, and the Face and Object Recognition Questionnaire. These criteria resulted in 294 web controls. The age restriction was imposed to avoid possible age-related decline on the CFMT (Bowles, McKone, Dawel, Duchaine, Palermo, Schmalzl, Rivolta, Wilson, & Yovel, 2009) and resulted in a sample with an age range identical to that of the DP group.

It is possible that some of the Richler et al. (2017) web participants have DP, given that the prevalence rate is believed to be as high as 2.5% (Kennerknecht, Grueter, Welling, Wentzek, Horst, Edwards, & Grueter, 2006). To avoid including potential DPs in our control sample, we removed any control participants who simultaneously scored *both* lower than 45 on the CFMT (Bowles et al., 2009) *and* higher than 43 on the administered face recognition questionnaire (one standard deviation above the group mean, indicating worse self-reported face recognition; Richler et al., 2017), as well as anyone who had a reaction time greater than three standard deviations above the control mean on either task. Eleven web participants were removed for a combined low CFMT and high face recognition questionnaire, 5 were removed due to a high

Running Head: OBJECT RECOGNITION PROSOPAGNOSIA

CFMT reaction time (>7.48 s; 3 SD above mean), and 4 were removed due to a high NOMT reaction time (>8.32 s; 3 SD above mean)³. Based on these criteria, 274 control participants were included from the original online sample.

One notable testing difference between the web control sample and DPs is that the testing session was briefer in the web sample compared to DPs (20 minutes vs. 3 hours), with fatigue effects that may have influenced performance in DPs not applicable to the web sample (Boksem, Meijman, & Lorist, 2005). While our in-lab controls performed the 3-hour long battery in the same order as the DPs (with the CFMT administered first and the NOMT administered approximately halfway through), the web controls performed the NOMT and CFMT after only one short questionnaire. This suggests that our in-lab control group better accounts for potential fatigue or order effects than does the web-based control group.

Informed consent was obtained for all participants prior to data collection according to the Declaration of Helsinki. Participants were compensated for their time at a rate of \$10 per hour. The study was approved by the VA Boston Healthcare System and Harvard Medical School Institutional Review Boards, and all study tasks were completed at either the VA Boston Healthcare System in Jamaica Plain or the Harvard Decision Science Lab.

Test Battery:

As our measure of face recognition ability, we used the original Cambridge Face Memory Test (CFMT), a widely used and highly validated measure of face memory (Duchaine & Nakayama, 2006a). ~~Because participants' CFMT scores were also used~~Using the CFMT as part of our diagnostic cut-off, ~~we naturally expect to see~~ naturally created large group differences in CFMT performance between the DP group and both control groups (DP vs. in-lab controls: $p < .001$, DP vs. web controls: $p < .001$). The benefit of using the CFMT as both a diagnostic measure and as our independent measure of face recognition ability is that we were able to create ~~two groups that are highly contrasted on~~ two groups with highly contrasting face recognition abilities, ~~between~~ between which we can compare respective object recognition abilities. To measure object recognition abilities, we used a novel object memory test (NOMT Ziggerins; Richler et al., 2017). The NOMT was modelled after the CFMT and mirrors its structure by

³ If the participants removed for potential DP are re-entered into the web sample ($N=294$), the web control group significantly differs from the DPs' mean CFMT score ($M_{DP}=38.30$, $SD_{DP}=3.72$, $M_{TD}=57.27$, $SD_{TD}=9.97$, $p < .001$; $d=2.52$), and does not significantly differ from the DPs' mean NOMT score ($M_{DP}=58.20$, $SD_{DP}=9.14$, $M_{TD}=61.07$, $SD_{TD}=7.97$, $p=.065$; $d=.33$).

Formatted: Font: (Default) Times New Roman

Formatted: Font: (Default) Times New Roman

presenting the viewer with a novel object shown from three separate viewpoints, which the viewer must subsequently select from among three similar objects (see Supplementary Figure 13). The trials repeat three consecutive times for each of the six target objects. After the initial 18 learning trials, all six learned objects are shown simultaneously for 20 seconds, followed by 30 test trials where the participant must select which from among three objects was one of the six they viewed previously. Finally, participants are shown another 20 seconds of the six learned objects followed by the remaining 24 test trials. For the final round of recognition trials, unlike the CFMT, the NOMT does not include visual noise intended to further obscure the stimuli.

All in-lab participants were tested on either a Lenovo laptop (34.5 x 19.5 cm display, 1920 x 1080 pixels, 60 Hz) or a Sony Vaio (34.29 x 19.05 cm, 1920 x 1080 pixels, 59 Hz).

Figure 1. Test phase for Novel Object Memory Test – Ziggerins. Adapted from Richler et al. (2017), used with permission from Dr. Isabel Gauthier.

Statistical Approach:

We performed independent-samples *t*-tests between the DP and control group for both CFMT and NOMT accuracy and reaction times to determine if DPs showed significant group-level differences on either task. To ensure a representative sample, we also compared the DPs'

Formatted: Centered

Formatted: Font: (Asian) Korean

Formatted: Indent: First line: 0 pi, Line spacing: single

CFMT and NOMT scores and reaction times to a larger web-based control group (Richler et al., 2017). We conducted additional Bayesian independent samples *t*-tests when the null hypothesis was confirmed. To check if the distribution of test scores and reaction times were equal between groups, we performed the Kolmogorov-Smirnov test of normality between the DP group and both control groups.- Subsequently, we performed Mann-Whitney U tests on the variables that showed unequal distribution across groups.

In addition to the group-level analyses, we also sought to determine whether DPs showed dissociations between tasks at the individual level. Using Crawford and colleagues' (2003) stringent definition of a classical dissociation, an individual would need to a) score significantly lower than controls on Task 1, b) show unimpaired performance on Task 2, and c) score significantly lower on Task 1 than on Task 2 ($p < 0.05$; two-tailed) to show a putative classical dissociation. If DPs show severely impaired performance on the CFMT (i.e., more than two standard deviations below the mean of the normative sample) while scoring normally on the NOMT (i.e., less than two standard deviations below the normative sample) with a significant difference between the two scores, then they could reasonably be classified as showing a dissociation between face and object recognition. We used the 30-person age- and gender-matched sample as our normative data. The in-lab sample was used in lieu of the web control sample due to its equivalence to the DPs in terms of the task battery and testing format. We also include results from the web sample in the supplementary materials. In addition to the dissociation analyses, we compared the prevalence of individual DP NOMT impairment to those of both the in-lab and web control groups using Fisher's exact test. To account for the difference in gender between the DP group and the web controls, we also divided the web controls by gender and compared individual DPs and controls to their respective gender on the NOMT (see supplementary material).

Results

Participants:

The DP group did not differ from the in-lab control group in either age ($M_{DP}=38.5$, $SD_{DP}=13.69$, $M_{ID}=40.03$, $SD_{ID}=11.73$, $p=.643$), gender ($t(57)=1.697$, $p=.095$), or education ($t(57)=1.926$, $p=.059$). The DP group did not differ from the web controls in in age ($M_{Web}=36.8$, $SD_{Web}=12.04$, $p=.464$), however, the DP group had a higher proportion of females ($N_{Female}=24$, 80.0%) than the web control group ($N_{Female}=160$, 58.2%; $t(298)=3.064$, $p=$

.004). Results separated by gender can be found in the supplementary materials (see Supplementary Figure 1). The web controls also had a lower average education level ($t(273) = 3.741, p < .001$). While ideally an education-matched web control group is preferred, studies have shown that face recognition is independent from both education and IQ (Wilmer, 2017). Similarly, education does not predict any additional variance in general memory performance outside of age (West, Crook, & Barron, 1992).

CFMT and NOMT Group-level Performance

First, we wanted to determine whether there were group-level differences on the CFMT and NOMT between the DPs and the matched in-lab controls. As expected, due to the qualification criteria, there was a significant difference between the two groups' CFMT scores ($t(58) = -13.631, p < .001; d = 3.52$), with the DP mean score far below that of the in-lab controls ($M_{DP} = 38.30, SD_{DP} = 3.72, M_{TD} = 59.77, SD_{TD} = 7.79$). However, there were no differences between DP ($M_{DP} = 58.20, SD_{DP} = 9.14$) and in-lab control group means ($M_{TD} = 57.00, SD_{TD} = 8.28$) for the NOMT ($t(58) = .533, p = .596; d = .14$) (see Figure 24). An additional Bayesian independent samples t -test revealed a Bayes factor of 4.52 in favour of the null hypothesis. CFMT reaction times were significantly slower for DPs than for in-lab controls ($t(47) = 2.4878, p = .017; d = .71$), whereas NOMT reaction times were not significantly different between the two groups ($t(58) = -.7658, p = .452; d = .20$) (see Table 1 and Supplementary Figure 35). To determine whether DP and control groups were sampled from similar versus different distributions, we performed a Kolmogorov-Smirnov test for each variable. The Kolmogorov-Smirnov test of normality showed that the distribution of DP scores and reaction times differed from in-lab controls for CFMT accuracy ($D(58) = 3.873, p < .001$), likely due to the constrained scores of the DP participants, but only showed a trend towards differing in CFMT reaction time ($D(47) = 1.3329, p = .059$), and did not differ between the groups for NOMT accuracy ($D(58) = .904, p = .388$) or NOMT reaction time ($D(58) = .7875, p = .586$). A confirmatory Mann-Whitney U test indicated that the difference between DP and in-lab control scores on the CFMT were significant for both accuracy ($U = .00, p < .001$) and reaction time ($U = 173.00, p = .017$) despite the variance in distribution.

We next compared the DP group results to the 274-person web sample. Similarly, we found that the DPs as a group showed worse performance on the CFMT ($t(302) = -22.244, p < .001; d = 2.74$), but only showed a trend towards a group-level difference on the NOMT ($t(302) =$

Formatted: Font: Italic

-1.874, $p = .062$; $d = .34$). A Bayesian independent samples t -test resulted in a Bayes factor of 1.31 in favour of the null hypothesis. Notably, the larger web sample differed from the DP group in terms of proportion of male versus female participants. When including gender as a covariate when comparing the NOMT accuracy between DP and web controls, the difference in scores failed to reach even a trend level of significance ($F(1, 297) = 2.5655, p = .111$). A figure of CFMT and NOMT results as a function of gender and group is included in the supplementary materials (see Supplementary Figure 1). NOMT reaction times for each group did not differ ($t(302) = .5109, p = .573$; $d = .10$), while DPs were significantly slower than controls for CFMT reaction times ($t(292) = 4.6879, p < .001$; $d = 1.26$) (see Table 1). A Kolmogorov-Smirnov test revealed that distributions of the DPs and web controls differed on CFMT accuracy ($D(302) = 4.763, p < .001$) and reaction times ($D(292) = 2.472, p < .001$) but not on NOMT accuracy ($D(302) = 0.9246, p = .371$) or reaction times ($D(302) = 1.154, p = .141$). A Mann-Whitney U test confirmed a significant difference between DP and web control scores on the CFMT ($U = 311.00, p < .001$).

Formatted: Font: Italic

Figure 21. Individual raw scores of out 72 for DP and in-lab and web-based control participants performing the Cambridge Face Memory Test and Novel Object Memory Test.

CFMT and NOMT Individual-level Performance

We next sought to determine whether any DPs evinced classical dissociations between their face and novel object recognition (Crawford et al., 2003). Using the in-lab control group as our normative data, 22 out of 30 (73.3%) DPs showed a putative classical dissociation between their NOMT and CFMT scores. These individuals scored in the unimpaired range on the NOMT (z-score > -2) but were significantly impaired on the CFMT (z-score < -2), and their performance on the CFMT was significantly lower than their performance on the NOMT (Crawford, Howell, & Garthwaite, 1998).

Out of the 8 DPs who did not evince a classical dissociation, 6 had NOMT z-scores > -1.7, and 7 had z-scores > -2. The remaining DP reached major impairment on the NOMT, scoring more than 2 standard deviations below the mean accuracy of the in-lab control group. Fisher’s exact test shows this prevalence rate of 1 out of 30 DPs to be comparable with the 10% (3/30; $p = .612$) rate of NOMT impairment found in the in-lab control group, as well as the 2.6% (7/274; $p = .569$) prevalence found in the web control sample. If the threshold for deficit is lowered to > 1.7 standard deviations below the mean (as used by Geskin and Behrmann), DPs show a prevalence rate of 6.7% (2/30), failing to differ from either the in-lab controls (10%, 3/30; $p = 1.00$) or the web controls (2.9%, 8/274; $p = .258$) according to Fisher’s exact test.

Table 1. NOMT and CFMT Performance

Measure	DP	In-lab TD	Web TD	p -values	
				In-lab	Web
CFMT Score	38.30 ± 3.72	59.77 ± 7.79	58.05 ± 9.48	.000**	.000**
NOMT Score	58.20 ± 9.14	57.00 ± 8.28	61.11 ± 7.96	.596	.062
CFMT Reaction Time (s)	5.18 ± 1.56	4.13 ± 1.37	3.52 ± 1.03	.017**	.000**
NOMT Reaction Time (s)	4.21 ± 1.40	4.49 ± 1.49	4.07 ± 1.20	.452	.573

Note. CFMT = Cambridge Face Memory Test. NOMT = Novel Object Memory Test. Mean ± standard deviation. *P*-values are derived from independent samples *t*-tests comparing DP and control test scores. **Statistically significant at $p < 0.05$.

Discussion

The goal of the current study was to compare DPs' novel object recognition ability to that of controls using a validated and reliable assessment matched in format to the CFMT. At the group level, DPs' NOMT accuracy and reaction time did not significantly differ from either the in-lab or larger web-based control groups. Only 1 out of the 30 DPs had a NOMT z-score < -2 and 2 out of 30 DPs had z-scores < -1.7 . These rates were not significantly different from the proportion of major and mild deficits found in either control group. Further, 22 out of the 30 DPs showed a classical dissociation between CFMT performance and NOMT performance, indicating severely impaired face recognition with preserved novel object recognition. ~~Six~~6 of the remaining 8 DPs had z-scores > -1.7 on the NOMT, suggesting that novel object recognition deficits are rare even in those not showing a classic dissociation. Together, these results observe a frequent dissociation across a large group of DPs, whereby face recognition impairment is associated with little to no impairment in a sensitive and well-validated test of novel object recognition that was designed to closely follow the structure of the CFMT. ~~Moreover, given the high correlation that was found previously between NOMT Ziggerins and other NOMTs (Richler et al., 2017), these results suggest that DPs as a population are broadly unimpaired in learning to recognise previously unfamiliar objects.~~ These results are consistent with those found with the 'blue objects' by Esins et al. (2014), with the added benefits of a test more comparable to the CFMT and an absence of floor effects. Considering these findings with recent studies showing DP impairments in familiar object recognition tasks (e.g., cars), this suggests that DPs have normal general object perception and recognition abilities but that some DPs may have a decreased capacity to benefit from their experience with familiar object categories.

According to Gauthier, NOMTs measure general visual abilities that reflect the capacity to "learn to match fingerprints (or) to perform radiological diagnosis" (Gauthier, 2018, p. 72). Based on this, we interpret DPs' unimpaired performance on the NOMT Ziggerins as reflecting normal domain-general aspects of object perception and recognition abilities for a class of objects that, in controls, correlates highly with performance on other novel objects. Notably, the abilities required to successfully perform the NOMT do not depend on access to associative representations (e.g., names, semantic information) or rely on object representations learned over time, but instead involve object perception and memory processes that are unrelated to one's level of category-specific object experience (Richler et al., 2017). DPs' normal performance on the NOMT Ziggerins likely reflects intact general abilities to perceive this class of objects as

well as to maintain *certain* object representations in memory over a delay. These intact mechanisms may at least partially underlie and support accurate perceptual and recognition performance with familiar object categories, particularly familiar objects associated with lower levels of expertise (Richler et al., 2017). Consistent with this, a large study found DPs to be essentially normal during sequential perceptual matching of familiar objects including hands and houses (while being impaired at face matching, Bate et al., 2019) and a smaller study found no difference between DPs' and controls' accuracy in simultaneous matching of both self and others' hands and feet (Malaspina, Albonico, & Daini, 2018). Further, larger studies of DPs have found that DP and control groups did not differ on familiar object recognition performance for bicycles (Cambridge Bicycle Memory Test; Barton et al., 2019), shoes (Stollhoff et al., 2011), shells (Esins et al., 2014), or flowers (Song et al., 2015).

DPs' normal group-level NOMT Ziggerins performance and previous reports of normal familiar object perception and recognition abilities (e.g., Bate et al., 2019) are also consistent with neural results comparing DPs and controls. When examining resting-state functional MRI connectivity in the visual 'object network', Song and colleagues found no differences between DPs and controls, though they found decreased connectivity amongst core face-selective regions (Song et al., 2015). Further, when analyzing an fMRI localiser task using short movie clips of faces, objects, and bodies, Jiahui et al. (2018) found normal familiar object selectivity (familiar objects vs. faces) in ventral temporal cortex in DPs, though interestingly they found reduced body selectivity in DPs (bodies vs. familiar objects) as well as reduced face selectivity (Jiahui, Yang, & Duchaine, 2018). Event-related potential studies have also shown normal DP responses to objects in the N170 (e.g., houses: Towler, Gosling, Duchaine, & Eimer, 2012), though not to bodies (Righart & de Gelder, 2007). Together, these neural findings are consistent with the current NOMT results and provide converging evidence that DPs have intact general object processing abilities that support the perception and recognition of familiar object categories with lower levels of expertise.

Despite evidence for normal object perception and recognition mechanisms in DPs as indexed by the NOMT, a significant number of studies have demonstrated impaired DP performance for certain familiar object categories, suggesting that DPs' object perception and recognition may not benefit as much from experience (Barton et al., 2019). This may particularly affect familiar objects with very high levels of experience, such as bodies and cars (McGugin et

al., 2012). For example, a study by Biotti et al. (2017) found that DPs have significant group-level deficits when perceptually matching bodies and cars. Further, Rivolta et al. (2017) showed that while DPs' body perception accuracy did not differ from controls', their reaction times were significantly slower (Rivolta, Lawson, & Palermo, 2017). With regard to object recognition, a large study found that DPs as a group had reduced car recognition performance (Gray et al., 2019) and a similar finding was observed in several smaller studies (e.g., Gerlach, Klargaard, & Starrfelt, 2016; Tanzer et al., 2016). Additionally, using a long-term recognition task with cars, Barton et al. (2019) found that DPs had reduced performance compared to controls when taking into account DPs' car expertise. Specifically, for every increment in car expertise as indexed by verbal semantic knowledge, DPs had only about half the gains in car recognition accuracy that were shown by healthy controls. Because object category expertise is rarely measured in DP studies, it is difficult to know if DPs' deficits are specific to object categories with very high levels of expertise such as cars and bodies or rather are more common across familiar object categories, perhaps explaining heterogeneous performance in less familiar object categories. It would be useful for future studies to examine object perception and recognition using categories ranging from less familiar to highly familiar while assessing DPs' and controls' category expertise (similar to Barton et al., 2019).

DPs' normal NOMT performance existing alongside deficits in expertise-dependent familiar object perception and recognition could have several explanations. First, studies have linked greater holistic perceptual processing to better recognition ability (e.g., DeGutis, Wilmer, Mercado, & Cohan, 2013) and familiar objects such as cars and bodies have shown to be processed in a more holistic manner than more novel objects, showing significant effects of inversion (cars: Rossion & Curran, 2010; bodies: Reed, Stone, Bozova, & Tanaka, 2003; Greebles in Greeble 'experts': Rossion, Gauthier, Goffaux, Tarr, & Crommelinck, 2002). Interestingly, DPs have shown to have similar, albeit modest inversion effects with cars compared to controls (Klargaard, Starrfelt, & Gerlach, 2018), but evidence of disrupted N170 inversion effects with bodies (Righart & de Gelder, 2007), suggesting potential deficits in holistic processing of familiar objects. These holistic processing deficits may contribute to DPs' poorer recognition memory for familiar objects by making familiar objects less perceptually distinct. An alternative explanation is that DPs are less able to make semantic or contextual associations with highly familiar objects. These associations could help with more elaborative

encoding of familiar objects and better subsequent recognition (Lockhart & Craik, 1990). This interpretation is consistent with the conceptualisation of developmental prosopagnosia as a disconnection syndrome between perceptual regions in the ventral temporal cortex and more anterior temporal regions involved in storing semantic knowledge (Avidan, Tanzer, Hadj-Bouziane, Liu, Ungerleider, & Behrmann, 2014).

The presence of deficits in some DPs for highly familiar objects while exhibiting normal NOMT performance has an interesting resemblance to the case of AW, the only reported case of selective developmental object agnosia (DOA; Germine, Cashdollar, Düzel, & Duchaine, 2011). AW showed severe impairments in recognition memory across numerous object categories, including doors, cars, guns, and tools, though not all categories, houses and sunglasses were unimpaired. Notably, AW was able to perform normally at a novel object recognition test, similar to DPs in the current study. AW's normal performance on novel object recognition while being impaired in recognizing certain categories of familiar objects parallels findings in DPs and suggests that DOA and DP (or at least some cases of DP) may involve similar mechanisms. This could be because these disorders co-occur or rather because they may disrupt overlapping mechanisms along the ventral visual/anterior temporal lobe processing stream (see Gray and Cook, 2018, for a discussion). To better understand potential similarities between DOA and DP, it would be useful for future studies to compare object perception and recognition of a broad array of familiar object categories (while taking into account object expertise) as well as assess performance using identical NOMTs.

It should be noted that though DPs have shown group-level deficits in perceiving and recognizing familiar objects, such as with cars and bodies (e.g., Gray et al., 2019; Barton et al., 2019), the proportion of individual DPs showing impairments are often only a small minority (z -scores < -2 : 4.3%; Gray et al., 2019; 16.7%; Barton et al., 2019; 22.0%, Geskin & Behrmann, 2017). Thus, the majority of DPs have normal familiar object perception and recognition abilities, even with highly familiar objects. Additionally, caution should be used when interpreting an individual's performance on any single familiar object category (as is often done in case analyses in DP studies), since even performance in unimpaired participants may be heterogeneous across familiar object categories, likely due to differences in experience and expertise. For example, Edward, a severe DP, was able to perform normally on multiple old/new tasks of familiar object recognition including horses, tools, and houses, and was able to

successfully gain expertise with novel objects in a Greebles training programme (Duchaine et al., 2006). However, he performed poorly on an old/new task with guns. Had Edward been only assessed on gun performance, his normal object perception, learning, and recognition abilities apart from guns would have been missed. Additionally, it is notable that even in the DOA AW, there were familiar object categories, including houses and sunglasses, for which her scores were completely normal. Thus, future studies with large samples of DPs employing perceptual and recognition tasks with a broad range of familiar object categories would be useful to determine the prevalence and heterogeneity of DPs' familiar object recognition difficulties.

Though the findings of the current study are compelling and offer a novel perspective on DPs' object processing abilities, this study has several limitations. First, we used only a single novel object recognition task. Although NOMTs may be more representative of category-general processes, ~~and NOMT Ziggerins correlates highly with the other NOMT measures (Richler et al., 2017)~~, it would have been better to test DPs with multiple NOMTs to better quantify this ability. ~~While NOMTs have shown to correlate highly with one another (Richler et al., 2017) and previous case studies have found normal DP performance with Greebles (Duchaine, Dingle, Butterworth, & Nakayama,~~

~~2004), further investigations would be useful to confirm the generalizability of normal NOMT performance in DPs. While NOMTs have been shown to correlate highly between one another (Richler et al., 2017) and case studies have found normal DP performance with Greebles (Duchaine, Dingle, Butterworth, & Nakayama, 2004), these findings may not generalise to DPs as a whole, and should be further investigated.~~ Second, the current study only assessed novel object recognition and it would be important for future studies to test both NOMTs and several categories of familiar object recognition as well as familiar object perceptual matching tasks (while factoring in category expertise) to more fully elucidate DPs' object processing abilities.

In sum, we show that as a group DPs perform equally as well as control subjects on a test of novel object memory, and that the proportion of DPs who are impaired was not significantly different from controls. This finding, combined with recent DP studies of familiar object recognition, suggests that object recognition deficits do not inevitably accompany face recognition impairments. Instead, they suggest that DPs' previously reported familiar object perception and recognition deficits stem from a decreased perceptual or mnemonic capacity to benefit from their experience with familiar object categories.

Acknowledgements

We want to thank our developmental prosopagnosic and control participants for completing our challenging battery of tasks.

Authors' Contributions

RF carried out data collection, participated in data analysis, and drafted the manuscript; JW provided additional data, designed and provided figures, and critically revised the manuscript; IX carried out data collection and assisted in drafting the manuscript; MV critically revised the manuscript; JD conceived of the study, designed and coordinated the study, and helped draft the manuscript.

Data Availability

Supporting data can be accessed at Dryad: <https://doi.org/10.5061/dryad.4b8gth95>

Funding

This study was funded by R01 from the National Eye Institute grant #RO1EY026057 awarded to JD. MV's effort was supported by a Senior Research Career Scientist award from the Clinical Science Research and Development Service, Department of Veterans Affairs. The contents of this manuscript do not represent the views of the U. S. Department of Veterans Affairs or the United States Government.

Competing Interests

The authors report no competing interests.

References

- APA Dictionary of Psychology. (n.d.). Retrieved February 6, 2020 from <https://dictionary.apa.org/recognition-memory>
- Avidan, G., Tanzer, M., Hadj-Bouziane, F., Liu, N., Ungerleider, L. G., & Behrmann, M. (2013). Selective Dissociation Between Core and Extended Regions of the Face Processing Network in Congenital Prosopagnosia. *Cerebral Cortex*, 24(6), 1565–1578. doi: 10.1093/cercor/bht007
- Baron-Cohen, S., Wheelwright, S., Skinner, R. *et al.* The Autism-Spectrum Quotient (AQ): Evidence from Asperger Syndrome/High-Functioning Autism, Males and Females,

Formatted: Font: Bold

Formatted: Font: (Default) Times New Roman

Formatted: Font: (Default) Times New Roman

Formatted: Font: (Default) Times New Roman

Formatted: Font: Bold

Field Code Changed

Formatted: Font: Times New Roman, 12 pt

Running Head: OBJECT RECOGNITION PROSOPAGNOSIA

Scientists and Mathematicians. *J Autism Dev Disord* **31**, 5–17 (2001).

<https://doi.org/10.1023/A:1005653411471>

- Barton, J. J. S., Albonico, A., Susilo, T., Duchaine, B., & Corrow, S. L. (2019). Object recognition in acquired and developmental prosopagnosia. *Cognitive Neuropsychology*, *36*(1-2), 54–84. doi: 10.1080/02643294.2019.1593821
- Bate, S., Bennetts, R. J., Tree, J. J., Adams, A., & Murray, E. (2019). The domain-specificity of face matching impairments in 40 cases of developmental prosopagnosia. *Cognition*, *192*, 104031. doi: 10.1016/j.cognition.2019.104031
- Behrmann, M., & Avidan, G. (2005). Congenital prosopagnosia: face-blind from birth. *Trends in Cognitive Sciences*, *9*(4), 180–187. doi: 10.1016/j.tics.2005.02.011
- Bentin, S., Deouell, L. Y., & Soroker, N. (1999). Selective visual streaming in face recognition. *NeuroReport*, *10*(4), 823–827. doi: 10.1097/00001756-199903170-00029
- Biotti, F., Gray, K. L., & Cook, R. (2017). Impaired body perception in developmental prosopagnosia. *Cortex*, *93*, 41–49. doi: 10.1016/j.cortex.2017.05.006
- Boksem, M. A., Meijman, T. F., & Lorist, M. M. (2005). Effects of mental fatigue on attention: An ERP study. *Cognitive Brain Research*, *25*(1), 107–116. doi: 10.1016/j.cogbrainres.2005.04.011
- Bowles, D. C., Mckone, E., Dawel, A., Duchaine, B., Palermo, R., Schmalzl, L., ... Yovel, G. (2009). Diagnosing prosopagnosia: Effects of ageing, sex, and participant–stimulus ethnic match on the Cambridge Face Memory Test and Cambridge Face Perception Test. *Cognitive Neuropsychology*, *26*(5), 423–455. doi: 10.1080/02643290903343149
- Crawford, J. R., Howell, D. C., & Garthwaite, P. H. (1998). Payne and Jones Revisited: Estimating the Abnormality of Test Score Differences Using a Modified Paired Samples t Test. *Journal of Clinical and Experimental Neuropsychology*, *20*(6), 898–905. doi: 10.1076/jcen.20.6.898.1112
- Crawford, J., Garthwaite, P., & Gray, C. (2003). Wanted: Fully Operational Definitions of Dissociations in Single-Case Studies. *Cortex*, *39*(2), 357–370. doi: 10.1016/s0010-9452(08)70117-5
- DeGutis, J., Wilmer, J., Mercado, R. J., & Cohan, S. (2013). Using regression to measure holistic face processing reveals a strong link with face recognition ability. *Cognition*, *126*(1), 87–100. doi: 10.1016/j.cognition.2012.09.004

Running Head: OBJECT RECOGNITION PROSOPAGNOSIA

Dennett, H. W., Mckone, E., Tavashmi, R., Hall, A., Pidcock, M., Edwards, M., & Duchaine, B. (2011). The Cambridge Car Memory Test: A task matched in format to the Cambridge Face Memory Test, with norms, reliability, sex differences, dissociations from face memory, and expertise effects. *Behavior Research Methods*, *44*(2), 587–605. doi: 10.3758/s13428-011-0160-2

Duchaine, B. C., Dingle, K., Butterworth, E., & Nakayama, K. (2004). Normal greeble learning in a severe case of developmental prosopagnosia. *Neuron*, *43*(4), 469–473. doi: 10.1016/j.neuron.2004.08.006

Duchaine, B., & Nakayama, K. (2006a). The Cambridge Face Memory Test: Results for neurologically intact individuals and an investigation of its validity using inverted face stimuli and prosopagnosic participants. *Neuropsychologia*, *44*(4), 576–585. doi: 10.1016/j.neuropsychologia.2005.07.001

Duchaine, B. C., & Nakayama, K. (2006). Developmental prosopagnosia: a window to content-specific face processing. *Current Opinion in Neurobiology*, *16*(2), 166–173. doi: 10.1016/j.conb.2006.03.003

Duchaine, B. C., Yovel, G., Butterworth, E. J., & Nakayama, K. (2006). Prosopagnosia as an impairment to face-specific mechanisms: Elimination of the alternative hypotheses in a developmental case. *Cognitive Neuropsychology*, *23*(5), 714–747. doi: 10.1080/02643290500441296

Duchaine, B., Yovel, G., & Nakayama, K. (2007). No global processing deficit in the Navon task in 14 developmental prosopagnosics. *Social Cognitive and Affective Neuroscience*, *2*(2), 104–113. doi: 10.1093/scan/nsm003

Esins, J., Schultz, J., Stemper, C., Kennerknecht, I., & Bülthoff, I. (2016). Face Perception and Test Reliabilities in Congenital Prosopagnosia in Seven Tests. *i-Perception*, *7*(1), 204166951562579. doi: 10.1177/2041669515625797

Esins, J., Schultz, J., Wallraven, C., & Bülthoff, I. (2014). Do congenital prosopagnosia and the other-race effect affect the same face recognition mechanisms? *Frontiers in Human Neuroscience*, *8*. doi: 10.3389/fnhum.2014.00759

Garrido, L., Duchaine, B., & Nakayama, K. (2008). Face detection in normal and prosopagnosic individuals. *Journal of Neuropsychology*, *2*(1), 119–140. doi: 10.1348/174866407x246843

Formatted: Font: Italic

Formatted: Font: Italic

Formatted: Font: (Default) Times New Roman, 12 pt

Formatted: Font color: Auto

Running Head: OBJECT RECOGNITION PROSOPAGNOSIA

- Gauthier, I. (2018). Domain-Specific and Domain-General Individual Differences in Visual Object Recognition. *Current Directions in Psychological Science*, *27*(2), 97–102. doi: 10.1177/0963721417737151
- Gerlach, C., Klargaard, S. K., & Starrfelt, R. (2016). On the Relation between Face and Object Recognition in Developmental Prosopagnosia: No Dissociation but a Systematic Association. *Plos One*, *11*(10). doi: 10.1371/journal.pone.0165561
- Germine, L., Cashdollar, N., Düzel, E., & Duchaine, B. (2011). A new selective developmental deficit: Impaired object recognition with normal face recognition. *Cortex*, *47*(5), 598–607. doi: 10.1016/j.cortex.2010.04.009
- Geskin, J., & Behrmann, M. (2017). Congenital prosopagnosia without object agnosia? A literature review. *Cognitive Neuropsychology*, *35*(1-2), 4–54. doi: 10.1080/02643294.2017.1392295
- Gray, K. L. H., Biotti, F., & Cook, R. (2019). Evaluating object recognition ability in developmental prosopagnosia using the Cambridge Car Memory Test. *Cognitive Neuropsychology*, *36*(1-2), 89–96. doi: 10.1080/02643294.2019.1604503
- Gray, K. L. H., & Cook, R. (2018). Should developmental prosopagnosia, developmental body agnosia, and developmental object agnosia be considered independent neurodevelopmental conditions? *Cognitive Neuropsychology*, *35*(1-2), 59–62. doi: 10.1080/02643294.2018.1433153
- Jiahui, G., Yang, H., & Duchaine, B. (2018). Developmental prosopagnosics have widespread selectivity reductions in category-selective areas. *Journal of Vision*, *18*(10), 916. doi: 10.1167/18.10.916
- Kennerknecht, I., Grueter, T., Welling, B., Wentzek, S., Horst, J., Edwards, S., & Grueter, M. (2006). First report of prevalence of non-syndromic hereditary prosopagnosia (HPA). *American Journal of Medical Genetics Part A*, *140A*(15), 1617–1622. doi: 10.1002/ajmg.a.31343
- Klargaard, S., Starrfelt, R., & Gerlach, C. (2017). The face-inversion effect in developmental prosopagnosia. *Journal of Vision*, *17*(10), 623. doi: 10.1167/17.10.623
- Lockhart, R. S., & Craik, F. I. M. (1990). Levels of processing: A retrospective commentary on a framework for memory research. *Canadian Journal of Psychology/Revue Canadienne De Psychologie*, *44*(1), 87–112. doi: 10.1037/h0084237

Running Head: OBJECT RECOGNITION PROSOPAGNOSIA

- Malaspina, M., Albonico, A., & Daini, R. (2018). Self-face and self-body advantages in congenital prosopagnosia: evidence for a common mechanism. *Experimental Brain Research*, *237*(3), 673–686. doi: 10.1007/s00221-018-5452-7
- McGugin, R., Richler, J., Herzmann, G., Speegle, M., & Gauthier, I. (2012). The contribution of general object recognition abilities to face recognition. *Journal of Vision*, *12*(9), 810–810. doi: 10.1167/12.9.810
- Reed, C. L., Bozova, S., Stone, V. E., & Tanaka, J. (2001). The body inversion effect. *PsychEXTRA Dataset*. doi: 10.1037/e537102012-763
- Richler, J. J., Wilmer, J. B., & Gauthier, I. (2017). General object recognition is specific: Evidence from novel and familiar objects. *Cognition*, *166*, 42–55. doi: 10.1016/j.cognition.2017.05.019
- Righart, R., & Gelder, B. D. (2007). Impaired face and body perception in developmental prosopagnosia. *Proceedings of the National Academy of Sciences*, *104*(43), 17234–17238. doi: 10.1073/pnas.0707753104
- Rivolta, D., Lawson, R. P., & Palermo, R. (2017). More than just a problem with faces: altered body perception in a group of congenital prosopagnosics. *Quarterly Journal of Experimental Psychology*, *70*(2), 276–286. doi: 10.1080/17470218.2016.1174277
- Rossion, B., & Curran, T. (2010). Visual Expertise with Pictures of Cars Correlates with RT Magnitude of the Car Inversion Effect. *Perception*, *39*(2), 173–183. doi: 10.1068/p6270
- Rossion, B., Gauthier, I., Goffaux, V., Tarr, M., & Crommelinck, M. (2002). Expertise Training with Novel Objects Leads to Left-Lateralized Facelike Electrophysiological Responses. *Psychological Science*, *13*(3), 250–257. doi: 10.1111/1467-9280.00446
- Shah, P., Gaule, A., Sowden, S., Bird, G., & Cook, R. (2015). The 20-item prosopagnosia index (PI20): a self-report instrument for identifying developmental prosopagnosia. *Royal Society Open Science*, *2*(6), 140343. doi: 10.1098/rsos.140343
- Song, Y., Zhu, Q., Li, J., Wang, X., & Liu, J. (2015). Typical and Atypical Development of Functional Connectivity in the Face Network. *Journal of Neuroscience*, *35*(43), 14624–14635. doi: 10.1523/jneurosci.0969-15.2015
- Susilo, T., & Duchaine, B. (2013). Advances in developmental prosopagnosia research. *Current Opinion in Neurobiology*, *23*(3), 423–429. doi: 10.1016/j.conb.2012.12.011
- Stollhoff, R., Jost, J., Elze, T., & Kennerknecht, I. (2011). Deficits in Long-Term Recognition

Running Head: OBJECT RECOGNITION PROSOPAGNOSIA

Memory Reveal Dissociated Subtypes in Congenital Prosopagnosia. *PLoS ONE*, 6(1).
doi: 10.1371/journal.pone.0015702

Tanzer, M., Weinbach, N., Mardo, E., Henik, A., & Avidan, G. (2016). Phasic alertness enhances processing of face and non-face stimuli in congenital prosopagnosia. *Neuropsychologia*, 89, 299–308. doi: 10.1016/j.neuropsychologia.2016.06.032

Torfs, K., Vancleef, K., Lafosse, C., Wagemans, J., & de-Wit, L. (2014). The Leuven Perceptual Organization Screening Test (L-POST), an online test to assess mid-level visual perception. *Behavior Research Methods*, 46(2), 472–487.

Towler, J., Gosling, A., Duchaine, B., & Eimer, M. (2012). The face-sensitive N170 component in developmental prosopagnosia. *Neuropsychologia*, 50(14), 3588–3599. doi: 10.1016/j.neuropsychologia.2012.10.017

Wegrzyn, M., Garlichs, A., Heß, R. W. K., Woermann, F. G., & Labudda, K. (2018). The hidden identity of faces: A case of lifelong prosopagnosia. doi: 10.31219/osf.io/dv7am

West, R. L., Crook, T. H., Barron, K.L. (1992). Everyday memory performance across the life span: Effects of age and noncognitive individual differences. *Psychology and Aging*, 7(1), 72–82.

Wilmer, J. B. (2017). Individual Differences in Face Recognition: A Decade of Discovery. *Current Directions in Psychological Science*, 26(3), 225–230. doi: 10.1177/0963721417710693

Appendix B

Dear Dr. Gliga,

Thank you for the invitation to further revise the manuscript “Evidence for normal novel object recognition abilities in developmental prosopagnosia”. In accordance with Royal Society Open Science guidelines, we have uploaded our de-identified data to Dryad. It can be accessed at https://datadryad.org/stash/share/06mAI-el7Wj4M9Pfs4DearipgsQhbyEME_CPXSIynfo, or with the DOI 10.5061/dryad.59zw3r25d once made public. This change is reflected in the section of the manuscript labeled “Data Availability”.

Thank you again for your comments, and for those of the reviewers.

Sincerely,
Regan Fry